# Spatial and temporal variability of the hydroxyl radical: Understanding the role of large-scale climate features and their influence on OH through its dynamical and photochemical drivers

Daniel C. Anderson[1,2], Bryan N. Duncan[2], Arlene M. Fiore[3], Colleen B. Baublitz[3], Melanie B. Follette-Cook[2,4], Julie M. Nicely[2,5], Glenn M. Wolfe[2]

[1] Universities Space Research Association, GESTAR, Columbia, MD, USA

[2] Atmospheric Chemistry and Dynamics Laboratory, NASA Goddard Space Flight Center, Greenbelt, MD

[3] Department of Earth and Environmental Sciences, Columbia University, Palisades, NY

[4] Morgan State University, GESTAR, Baltimore, MD

[5] Earth System Science Interdisciplinary Center, University of Maryland, College Park, College Park, MD

*Correspondence to: Daniel C. Anderson (daniel.c.anderson@nasa.gov)*

## Abstract

The hydroxyl radical (OH) is the primary atmospheric oxidant, responsible for removing many important trace gases, including methane, from the atmosphere. Although robust relationships between OH drivers and modes of climate variability have been shown, the underlying mechanisms between OH and these climate modes, such as the El Niño Southern Oscillation (ENSO), have not been thoroughly investigated. Here, we use a chemical transport model to perform a 38-year simulation of atmospheric chemistry, in conjunction with satellite observations, to understand the relationship between tropospheric OH and ENSO, Northern Hemispheric modes of variability, the Indian Ocean Dipole, and monsoons. Empirical orthogonal function (EOF) and regression analyses show that ENSO is the dominant mode of global OH variability in the tropospheric column and upper troposphere, responsible for approximately 30% of the total variance in boreal winter. Reductions in OH due to El Niño are centered over the tropical Pacific and Australia and can be as high as 10 - 15% in the tropospheric column. The relationship between ENSO and OH is driven by changes in nitrogen oxides in the upper troposphere and changes in water vapor and $O^1D$ in the lower troposphere. While the correlations between monsoons or other modes of variability and OH span smaller spatial scales than for ENSO, regional changes in OH can be significantly larger than those caused by ENSO. Similar relationships occur in multiple models that participated in the Chemistry Climate Model Initiative (CCMI), suggesting that the dependence of OH interannual variability on these well-known modes of climate variability is robust. Finally, the spatial pattern and $r^2$ values of correlation between ENSO and modeled OH drivers – such as carbon monoxide, water vapor, lightning, and to a lesser extent, $NO_2$ – closely agree with satellite observations. The ability of satellite products to capture the relationship between OH drivers and ENSO provides an avenue to an indirect OH observation strategy and new constraints on OH variability.

## 1.0    Introduction

The hydroxyl radical (OH), the atmosphere's primary oxidant, removes many trace gases that affect composition and climate. Despite its central role in atmospheric chemistry, the spatiotemporal distributions of OH concentrations are poorly constrained, often confounding interpretation of observed variations and trends of important atmospheric constituents. For example, there are several plausible explanations of the observed fluctuations in the global burden of atmospheric methane ($CH_4$), the second-most important anthropogenic greenhouse gas. Explanations include variations and trends in both emissions and oxidation of methane (Prather and Holmes, 2017;Rigby

et al., 2017;Turner et al., 2017).  Better constraints on OH and its dynamical and photochemical
drivers are needed to improve confidence in our interpretation of recent methane trends and to
project future climate in response to changes in emissions and composition.
Observational limitations and chemistry climate model disagreement pose challenges to advancing
our understanding of the spatiotemporal variability in OH. There are few direct *in situ* OH
observations, on local, regional, and global scales (Stone et al., 2012) as OH is both highly reactive,
with a lifetime of ~1 s in the free troposphere (Mao et al., 2009), and low in concentration, on the
order of $10^6$ molecules/$cm^3$.  Recent work has demonstrated that formaldehyde, a longer-lived
species (hours) whose chemical production in the remote troposphere is dominated by $CH_4$
oxidation, shows promise for inferring variability in OH columns over the remote atmosphere
(Wolfe et al., 2019).  In models of atmospheric chemistry and transport, OH can vary widely, with
differences in global methane lifetime, a proxy for OH abundance, between 45 and 80% among
models in inter-comparison projects (e.g., Voulgarakis et al., 2013;Nicely et al., 2017;Zhao et al.,
64  2019).
Analysis of the factors causing inter-model differences in the tropospheric OH burden is
challenging, as causation is difficult to prove with a species so tightly coupled to a multitude of
chemical and meteorological processes.  Primary OH production occurs through photolysis of $O_3$
followed by reaction with water vapor ($H_2O_{(v)}$), while secondary production is often regulated by
nitrogen oxides ($NO_X$ = NO + $NO_2$) through the reaction of the hydroperoxyl radical ($HO_2$) with NO.
Globally, CO and $CH_4$ are the primary sinks, although other species, particularly volatile organic
compounds (VOCs), can be important regionally.  However, attributing OH variability remains
challenging, with different models showing widely ranging responses in OH to changes in these
drivers, particularly to $NO_X$ and humidity (Wild et al., 2020).
These chemical and radiative drivers of OH variability are in turn partially regulated by large-scale
dynamical features, such as the El Niño Southern Oscillation (ENSO), monsoons, and modes of
Northern Hemispheric (NH) variability (*e.g.* the North Atlantic Oscillation (NAO)), through changes
in transport and emissions.  Oman et al. (2011) and Oman et al. (2013) used satellite observations
and chemistry climate models to show that the horizontal and vertical distributions of tropospheric
ozone are significantly modulated by ENSO, most prominently through the manifestation of a dipole
pattern over southeast Asia and the tropical western Pacific.  Sekiya and Sudo (2012) found similar
results with the CHASER chemical transport model, along with strong relationships between ozone
variability and the Indian Ocean Dipole (IOD), the Arctic Oscillation, and the Asian winter monsoon.
ENSO events can also change $CH_4$ emissions from wetlands (Zhang et al., 2018), lightning NO
production (Murray et al., 2013;Murray et al., 2014;Turner et al., 2018), and CO emissions from
biomass burning (Duncan, 2003a;Duncan, 2003b;Rowlinson et al., 2019).  In addition to this
biomass burning relationship with ENSO, Buchholz et al. (2018) also noted relationships between
CO in tropical fire regions and the IOD as well as with the Tropical South Atlantic and Southern
Annular modes.  Relationships between the Madden-Julian Oscillation (MJO) and variability of
tropical ozone (Tian et al., 2007;Ziemke et al., 2015), $H_2O_{(v)}$ (Myers and Waliser, 2003), and CO
(Wong and Dessler, 2007) have also been shown.  Finally, climate modes can alter the long range
transport of CO to the Arctic, through increased outflow from Europe (Li et al., 2002;Creilson et al.,
2003;e.g. Duncan, 2004) and Asia (Fisher et al., 2010) for the NAO and ENSO, respectively.
Despite the strong linkages between these dynamical features and OH drivers, there is little
research on the relationship between these processes and OH itself.  Turner et al. (2018) used a
6000-year simulation with free running dynamics to suggest that ENSO is the dominant mode of OH
variability at decadal timescales, mainly through its effects on lightning NO emissions.  Their study,
however, held most forcings and emissions, including greenhouse gas concentrations and biomass
burning, to 1860 conditions. Emissions of lightning NO, dust, and dimethyl sulfide were allowed to
respond to model meteorology. During the 1997/98 ENSO event, increases in CO from biomass
burning led to decreases in OH of 9% on the global scale (Rowlinson et al., 2019) and up to 20%
over the Indian Ocean (Duncan, 2003a). Using inversions of observations of methyl chloroform to
estimate OH concentrations, Prinn et al. (2001) found OH to be lower during ENSO years,
suggesting this could be linked to reduced UV radiation near the surface due to increased cloud
coverage. As with ENSO, modeling studies have shown that the Asian monsoon increases OH
concentrations in the upper troposphere (UT) through increased lightning NO production, despite
increases in convectively lofted OH sinks, particularly CO (Lelieveld et al., 2018).
Here, we examine how OH and related chemical and radiative factors vary with known modes of
climate and atmospheric variability. Using correlation analysis, we compare the relationship
between ENSO and tropospheric column OH from the MERRA-2 GMI (Modern-Era Retrospective
analysis for Research and Applications Global Modeling Initiative) setup of the NASA Goddard Earth
Observing System (GEOS) Chemistry Climate Model (GEOSCCM) (Strode et al., 2019) and four
models that participated in the joint International Global Atmospheric Chemistry
(IGAC)/Stratosphere-troposphere Processes And their Role in Climate (SPARC) Chemistry Climate
Model Initiative (CCMI) (Morgenstern et al., 2017). After evaluating these relationships from the
MERRA2 GMI model with *in situ* and satellite observations, we explore further the relationship
between OH, its precursors, and ENSO. Finally, we expand the analysis to include not only ENSO
but also other modes of internal climate variability.
**2.0     Methods**
In this section, we outline the methodology used to understand the relationship between OH and
large-scale dynamical drivers. First, we describe the analysis methods used in Section 2.1. In
Sections 2.2 and 2.3, we describe the relevant details of the MERRA2 GMI and CCMI simulations,
respectively.
**2.1     Description of Analysis Methods**
Because the factors driving OH concentrations and interannual variability are altitude dependent,
we divide the atmosphere into 4 layers: the surface to the top of the PBL (PBL), from the top of the
PBL to 500 hPa (Lower Free Troposphere: LFT), between 500 and 300 hPa (Middle Free
Troposphere: MFT), and from 300 hPa to the tropopause (Upper Free Troposphere: UFT). Output
from each model has been vertically averaged to these layers on a seasonal basis. In addition, we
also examine the tropospheric column.
To help determine the relationship between the modes of climate variability and photochemical
and meteorological variables archived by the various models, we regress model output against
different climate indices. To perform the regression, we first detrend the output on a monthly
basis, removing any linear trend from each variable over the 1980 to 2018 period to account for
changes in the background value. We then regress the model variable against a specific climate
index (*e.g.* ENSO index) for 1980 to 2018. We perform these regressions on each grid cell for each
of the 4 layers as well as for the tropospheric column. In the results below, we only include
regressions where the Pearson correlation coefficient (r) exceeds 0.5, unless otherwise indicated.
Using other methods to define significance of a regression, such as a two-tailed student t test with p
values less than 0.05, does not significantly alter the results.

Climate features considered here include ENSO, the IOD, several Northern Hemispheric
atmospheric modes of variability, and various monsoons.  We use monthly values of the ENSO
multivariate index (MEI) (Wolter and Timlin, 2011) obtained from https://psl.noaa.gov/enso/mei
and averaged to seasonal time scales.  Here, ENSO-related events are defined according to the
seasonally averaged MEI, where MEI > 0.5 is an El Niño event, MEI < -0.5 is a La Niña event, and an
MEI value between 0.5 and -0.5 is a neutral event.  For the Indian Ocean Dipole, we used the Dipole
Mode Index (DMI) obtained from  https://psl.noaa.gov/gcos_wgsp/Timeseries/DMI/.  Northern
Hemispheric modes considered are the NAO, the East Atlantic Pattern (EA), the Pacific North
American Pattern (PNA), the East Atlantic Western Russian Pattern, the Scandinavian Pattern, the
West Pacific Pattern, the East Pacific North Pacific Pattern, and the Tropical Northern Hemisphere
Pattern.  Indices for the NH modes were taken from the NOAA Climate Prediction Center (available
online at https://www.cpc.ncep.noaa.gov/data/teledoc/telecontents.shtml) and were determined
from a rotated principal component analysis of the 500 hPa geopotential height of the National
Center for Environmental Prediction Reanalysis.
The MERRA2 GMI (Section 2.2) and CCMI models (Section 2.3) included here are constrained or
nudged to reanalyses data (MERRA, MERRA2, JRA-55, and the ERA-interim) which assimilate
observed meteorology.  The meteorological variables used to calculate the DMI and MEI, including
sea surface temperature, sea level pressure, and zonal and meridional winds, agree well or are
identical among the different reanalyses (Orbe et al., 2020;Bosilovich et al., 2015).  Thus, climate
modes in these models correspond to the NOAA indices.  Likewise, indices for the NAO calculated
from surface pressure from the models correlate well ($r^2$ of 0.79 or greater) with the NAO index
calculated by NOAA.
**Table 1:** Summary of the climate modes and monsoons considered in this work.  The index used to
characterize the mode, as well as the source of the index, is also indicated.

| Mode Type | Index | Mode Type | Index |
|---|---|---|---|
| El Niño Southern Oscillation | Multivariate ENSO Index (NOAA) | North Atlantic Oscillation | EOF of geopotential height at 500 mbar from NCEP reanalysis (NOAA) |
| Indian Ocean Dipole | Dipole Mode Index (NOAA) | East Atlantic | |
| Asian Monsoon | Model-specific index calculated from the difference of zonal winds in monsoon specific regions | Pacific North American | |
| South American Monsoon | | East Atlantic Western Russian | |
| North American Monsoon | | Scandinavian | |
| South African Monsoon | | West Pacific | |
| North African Monsoon | | East Pacific North Pacific | |
| Australian Monsoon | | Tropical Northern Hemisphere | |
| Western North Pacific Monsoon | | | |

Monsoons included in this analysis are the Asian, South American, North American, South African,
North African, Australian, and the Western North Pacific.  We calculate the monsoon index for each
model used in this study based on the definitions of Yim et al. (2013), where the index is defined by
the difference of zonal winds at 850 hPa between two, monsoon-specific regions.  See Table 2 in
Yim et al. (2013) for more details.  Because the MERRA2 GMI and CCMI models included here are
constrained or nudged to different reanalyses, the calculated monsoon index varies among the
models, although the indices of a given monsoon from each model are highly correlated with one
another (generally $r^2 > 0.9$).  Table 1 summarizes the climate modes and monsoons as well as the
corresponding indices used here.
In addition to regression analysis, we also performed an empirical orthogonal function (EOF)
analysis for tropospheric column OH (TCOH) and separately for each of the four layers described
above.  EOF analysis allows for the statistical determination of the spatial modes of OH variability
and their variation with time without *a priori* knowledge of the controlling mechanisms (e.g.,
Barnston and Livezey, 1987).  To perform the analysis, OH fields for each grid box were detrended
by subtracting a linear fit to the time series over the 1980 to 2018 period to account for changes in
background associated with long-term trends in OH.  We report here only the first and second EOFs
and their associated principal component time series as none of the other EOFs correlated spatially
or temporally with any of the modes of climate variability discussed here.

## 2.2    MERRA-2 GMI Simulation Description

To understand the interannual variability of OH, we use the MERRA-2 GMI (Modern-Era
Retrospective analysis for Research and Applications Global Modeling Initiative) simulation,
publicly available at https://acd-ext.gsfc.nasa.gov/Projects/GEOSCCM/MERRA2GMI/.  This is a run
of the GEOSCCM model (Strode et al., 2019) constrained to meteorology from MERRA-2 (Gelaro et
al., 2017) that uses the GMI chemical mechanism (Duncan et al., 2007;Oman et al., 2013;Gelaro et
al., 2017).  The GMI chemical mechanism includes approximately 120 species and 400 reactions,
characterizing the photochemistry of the troposphere and stratosphere.  The model was run from
1980 to 2018 at a resolution of c180 on the cubed sphere, equivalent to approximately 0.625°
longitude × 0.5° latitude, with 72 vertical levels.  The model was run in a replay mode (Orbe et al.,
2017) and constrained to temperature, pressure, and winds from MERRA-2.  Model output is
available at daily and monthly resolutions, with hourly output available only for some local satellite
overpass times.  All data used in this work is monthly averaged unless otherwise indicated.

Anthropogenic emissions are from the Measuring Atmospheric Composition and Climate mega City
(MACCity) inventory (Granier et al., 2011) for 1980 – 2010, and then from the Representative
Concentration Pathway 8.5 (RCP8.5) scenario for 2011 – 2018.  Biomass burning emissions are
from the Global Fire Emissions Database (GFED) 4s inventory starting in 1997 (Giglio et al., 2013).
Biomass burning emissions from before 1997 are calculated from scale factors derived from
aerosol index data from the Total Ozone Mapping Spectrometer (TOMS) instrument, as described in
Duncan (2003).  Biogenic emissions are calculated online using the method described in Guenther
et al. (1999) and Guenther et al. (2000), an early form of the Model of Emissions of Gases and
Aerosols from Nature (MEGAN).  A known high bias in isoprene emissions from MEGAN (e.g., Wang
et al., 2017), could exacerbate low modeled OH in regions dominated by biogenic VOC emissions.
Lightning NO emissions are based on the cumulative mass flux (Allen et al., 2010), with constraints
from the Lightning Imaging Sounder (LIS)/Optical Transient Detector (OTD) v2.3 climatology (Cecil
et al., 2014).  Total, global lightning NO emissions are scaled to be 6.5 Tg N/year for each year of the
simulation, although emissions demonstrate significant interannual variability on the local
scale.  For example, over the tropical Pacific, an area we will investigate throughout this paper, peak
emissions are 1.5 times higher than minimum emissions over the time period studied here (Fig. S1).
Methane concentrations are specified at the surface for 4 different latitude bands (90°S - 30°S, 30°S
- 0°, 0° - 30°N, 30°N - 90°N) at monthly resolution and advected throughout the troposphere.
Methane data are from the NOAA Global Monitoring Division (GMD) surface network (Dlugokencky
et al., 1994) and monthly values are interpolated from annual means.
Because $CH_4$ is specified as a boundary condition, the model does not capture feedbacks (e.g.,
wetland or wildfire emissions) between $CH_4$ emissions and climate modes beyond the extent to
which these manifest in the observed methane surface concentrations.  ENSO, for example, is
known to affect atmospheric $CH_4$ concentrations through changes in emissions from wetlands
(Zhang et al., 2018;Melton et al., 2013) and biomass burning (Worden et al., 2013), although there
is uncertainty in the magnitudes of these effects (Melton et al., 2013).  On the global scale, however,
these ENSO-induced changes in emissions do not significantly perturb background $CH_4$.  For
example, during the 1997/98 ENSO event, one of the largest on record, $CH_4$ grew at a rate of
approximately 15 ppbv/yr on top of a background on the order of 1700 ppbv (Nisbet et al., 2016).
Because of this small perturbation and the dominance of CO as the primary OH sink over much of
the globe (see Section 5.0), it is unlikely that the relationship between climate modes and OH would
differ significantly with the inclusion of direct methane emissions in the simulation.
**2.3     IGAC/SPARC Chemistry Climate Model Initiative (CCMI) Phase 1 Model Simulations**
To place the results from MERRA2 GMI in the context of other models, we compare our simulation
with those from CCMI.  The CCMI was conducted to help assess the ability of a suite of models to
address various aspects of atmospheric chemistry, including trends in tropospheric ozone and the
controlling mechanisms of OH (Morgenstern et al., 2017).  Output from these models have already
been used to assess various aspects of tropospheric OH (Zhao et al., 2019;Nicely et al., 2020), HCHO
(Anderson et al., 2017), $O_3$ (Revell et al., 2018;Dhomse et al., 2018) and meteorological variables
(Orbe et al., 2020).  Modeling groups conducted multiple runs, including a forecast scenario to 2100
and two hindcast scenarios, one with free-running meteorology and one, the specified dynamics
(SD) scenario, in which models were either nudged to meteorological reanalyses or run as chemical
transport models (Orbe et al., 2020).
We perform a similar analysis as with MERRA2 GMI with four models that performed the CCMI SD
run.  We use the SD run, which spanned the years 1980 – 2010, instead of the other scenarios to
allow for more direct comparison among the CCMI models as well as with MERRA2 GMI and
observations from satellite.  We include only models that output data for all years between 1980
and 2010 and that have non-methane hydrocarbon chemistry in their chemical mechanisms.
Models used here are WACCM (Solomon et al., 2015), CHASER (MIROC-ESM) (Watanabe et al.,
2011), a setup of EMAC with 90 vertical levels (EMAC) (Jöckel et al., 2016), and MRI-ESM1r1
(Yukimoto et al., 2012).  We omit CAM4Chem and a different setup of EMAC with 47 vertical levels
because results for those models are essentially identical to WACCM and EMAC90, respectively.
EMAC90 and CHASER were nudged to the ERA-interim reanalysis, WACCM to the MERRA
reanalysis, and MRI to the JRA-55 reanalysis.  Sea surface temperatures (SST) and sea ice were
prescribed in each model with the Hadley SST dataset.  Anthropogenic emissions were from the
MACCity inventory, while lightning $NO_X$ was calculated online using model-specific
parameterizations.  Biomass burning emissions are from Granier et al. (2011), which incorporate a
modified version of the RETRO inventory from 1980 – 1996 and GFEDv2 from 1997 – 2010 and are
based on Lamarque et al. (2010).  Monthly-averaged CO emissions from this inventory in Indonesia,
where biomass burning emissions are strongly affected by ENSO (e.g., Duncan, 2003a), are highly
correlated ($r^2$ = 0.79) in time with the GFED 4s inventory used in the M2GMI simulation. Likewise,
monthly-averaged CO emissions over Indonesia from the two inventories agree within 35%, on
average.  Further model details can be found in Orbe et al. (2020), Morgenstern et al. (2017), and
references therein.
As with the MERRA2 GMI analysis, we use monthly averaged output.  For layer averaging, only
EMAC90, WACCM, and MRI output a tropopause height, while no models output PBL height.  To
calculate the tropopause height for CHASER, we used the relationship between $O_3$ and CO as
described in Pan et al. (2004).  PBL height for all models was determined from the bulk Richardson
number (Seibert et al., 2000).
**3.0 MERRA2 GMI Simulation Evaluation**
While there has been some evaluation of the MERRA2 GMI simulation (Ziemke et al., 2019;Strode et
al., 2019), species in the simulation relevant to this study have not been investigated.  As a result,
we evaluate MERRA2 GMI using *in situ* observations of OH and related species as well as remotely
sensed observations of OH drivers in order to understand the effect any model biases could have on
our results.  In Section 3.1, we use *in situ* observations from the first two deployments of the
Atmospheric Tomography (ATom) campaign to evaluate OH and CO over the remote Pacific and
Atlantic Oceans.  In Section 3.2, we also compare output to satellite observations of CO, $H_2O_{(v)}$, and
$NO_2$ to evaluate the model over larger temporal and spatial scales.
**3.1     Evaluation of MERRA2 GMI with *in situ* Observations**
During the ATom campaign, a suite of air quality and climate relevant trace gases and aerosols were
measured throughout the remote Pacific and Atlantic.  During each of the deployments, aircraft
transected the Pacific from Alaska to New Zealand, went around Tierra del Fuego, and travelled
north over the Atlantic to Greenland.  Each flight consisted of a series of ascents and descents
allowing for vertical profiling across most latitudes of the remote Pacific and Atlantic Oceans.  The
combination of the flight track and the repetition across seasons provided unprecedented sampling
of many trace gases, including OH.  As part of the ATom campaign, a limited subset of species,
including OH and CO, from the MERRA2 GMI simulation were output hourly for the duration of
ATom1 (July – August 2016) and ATom2 (January – February 2017) only, allowing for direct
comparison to the *in situ* observations.  Only daily or longer resolution output is available for the
other deployments, and, as a result, we focus our analysis on these first two deployments.

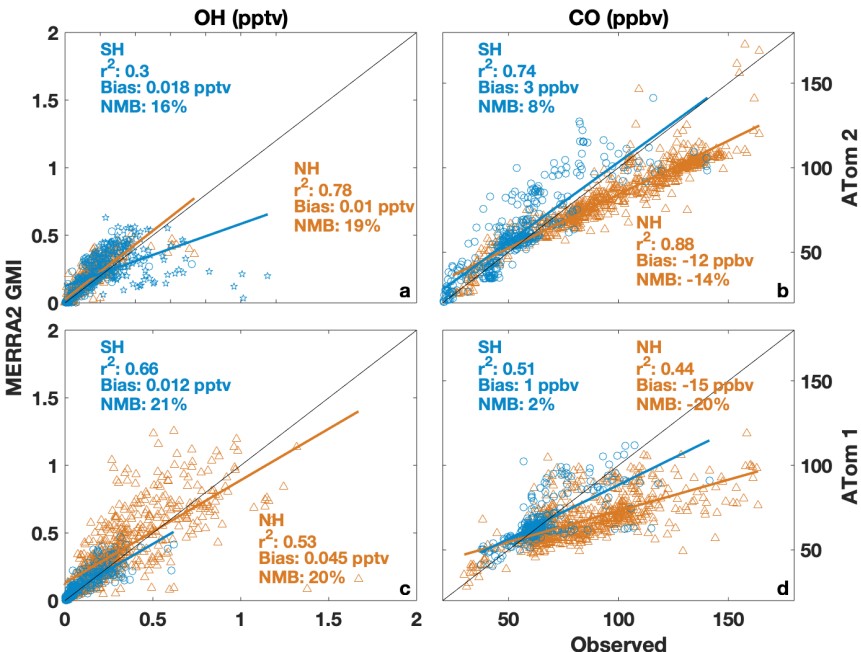

*Figure 1: Regression of observed OH (left) and CO (right) from ATom 2 (boreal winter 2017; top row) and ATom 1 (boreal summer 2016; bottom row) against hourly output from MERRA2 GMI interpolated to the ATom flight track. Data from the Southern (blue circles) and Northern (orange triangles) Hemisphere are shown, along with the r², bias, and normalized mean bias (NMB) for each hemisphere. Observations and model output have been filtered for biomass burning influence. Observations of continental outflow from New Zealand and South America from ATom 2 are indicated by blue stars.*

Observations used here include OH (Brune et al., 2020) and CO (Santoni et al., 2014), with $2\sigma$ uncertainties of 35% and 3.5 ppbv, respectively. Data have been averaged to a 5-minute time base and filtered for biomass burning influence, defined as times when concentrations of HCN and CO are both above the 75th percentile for the individual ATom deployments. We omit the biomass burning influenced parcels because small differences in measured and modeled winds could result in misplacement of modeled biomass burning plumes, resulting in unrealistically large differences in OH. Inclusion of the biomass burning influenced parcels does not significantly change the model bias but does degrade the correlation. For comparison of the observations to MERRA2 GMI, hourly data were output by the model and then bilinearly interpolated in the horizontal and linearly interpolated in time and in the vertical to the *in situ* observation time and location.

MERRA2 GMI has a OH high bias of approximately 20% (Fig. 1a) when compared to observations from ATom 2. A regression of measured and modeled OH shows moderate to high correlation in both the Southern Hemisphere (SH) and NH, with $r^2$ values of 0.30 and 0.78, respectively. Normalized Mean Biases (NMB) relative to the observations is within measurement uncertainty in both the NH (19%) and SH (16%), with nearly identical high biases during the summer deployment of ATom1 (Fig. 1c). The comparatively poorer model performance for OH in the SH is being driven by continental outflow from South America and New Zealand. When data from these regions are omitted (Fig. 1a, blue stars), the correlation for the SH increases to 0.63 and the NMB is 22%. The limited model output at hourly resolution does not allow for a determination of the cause of this disagreement in continental outflow regions. In the case of South America, however, a known high bias in modeled isoprene, resulting in extremely low OH over the Amazon, is consistent with the disagreement between the simulation and observations.

Agreement between observed and modeled CO shows a strong hemispheric dependence, with a
NMB of -14% in the NH (i.e., the model is lower than observations by 14%) and 8% in the SH during
ATom 2, although both hemispheres have a strong correlation ($r^2$ > 0.7). While agreement in the SH
improves in ATom 1, with a NMB of 2% (Fig. 1d), the model underestimate in the NH is even more
pronounced (NMB = -20%). This NH low bias in CO is a well-known problem in global chemistry
models (e.g. Naik et al., 2013;Stein et al., 2014;Travis et al., 2020) and could be a contributing factor
in the overestimate in OH, as CO is the dominant global OH sink.
Comparison of the MERRA2 GMI simulation to *in situ* observations demonstrates that the model
captures the spatial variability of OH and its predominant global sink, CO, in the remote atmosphere
during both the NH summer and winter, with the exception of OH off the coast of South America
and New Zealand. The poorer agreement between measured and modeled OH in regions of fresh,
continental outflow suggests that modeled relationships between climate modes and OH in these
regions might be more uncertain than in the remote atmosphere. This lack of agreement does not
significantly affect the results discussed in this work, as the majority of the relationships found
between OH and modes of climate variability discussed in Sections 4 and 5 are centered in the
remote atmosphere.
**3.2     Evaluation of MERRA2 GMI with Satellite Observations**
While there are no remotely sensed observations of tropospheric column OH (TCOH), there are
satellite observations of OH drivers. Comparing these observations to MERRA2 GMI allows for
model evaluation over larger spatial and temporal scales than with ATom. Satellite data used here
include tropospheric CO columns from the Measurement Of Pollutants In The Troposphere
(MOPITT) instrument, $H_2O_{(v)}$ from the Atmospheric Infrared Sounder (AIRS), and tropospheric $NO_2$
from the Ozone Monitoring Instrument (OMI). AIRS is on the Aqua satellite, with a daily, local
overpass time of approximately 13:30. We use the monthly averaged, Level 3, Version 6 standard
physical retrieval (Susskind et al., 2014) from 2003 to 2018. For MOPITT CO on the Terra satellite,
we use the Level 3, V008 retrieval that uses both near and thermal infrared radiances (Deeter et al.,
2019) from 2001 to 2018. MOPITT has a daily, local overpass time of approximately 10:30. Both
satellite products have a global horizontal resolution of 1° × 1°. We also use the OMI $NO_2$ Version 4,
Level 3 product (Lamsal et al., 2021) from 2005 to 2018. Data have been regridded to 1° × 1°
horizontal resolution. OMI is located on the Aura satellite and, as with AIRS, has a local overpass
time of approximately 13:30.
For comparison of the satellite retrievals to MERRA2 GMI, we use monthly fields of the model
variables output at the satellite overpass time. For CO, where averaging kernel and *a priori*
information are available for the Level 3 MOPITT data, we convolve the model output with these
variables so that direct comparison between satellite and model are possible. While shape factors
and scattering weights for the OMI $NO_2$ retrieval are unavailable for the Level 3 data, shape factors
for the OMI $NO_2$ retrieval are determined from a similar setup of the GEOSCCM model, also
employing the GMI chemical mechanism and MERRA2 meteorology. Applying the satellite shape
factors to the simulation discussed here would therefore not result in significant changes in the
modeled $NO_2$. Finally, for AIRS $H_2O_{(v)}$, averaging kernel information was unavailable for the Level 3
data, so numerical comparisons between satellite and model should be regarded as more
qualitative than quantitative.
When compared to MOPITT in boreal winter (*i.e.,* DJF), tropospheric column CO from MERRA2 GMI
(Fig. 2, first column) shows similar results to that found through comparison to the *in situ*
observations, namely a low bias in the NH (9%) and high bias in the SH (7%). Differences over the
tropical Pacific, an area that will be shown later to have a strong relationship between ENSO and
OH, are generally less than 10%, while a noticeable high bias exists over parts of South America.
Results for June – August (JJA) are spatially similar (Fig. 3), with a NH low bias of 20% and
overestimates of column CO, averaging 45%, in the SH.  These areas of high bias over South America
likely result from the high bias in isoprene emissions, as discussed in Section 2.2, that would lead to
unrealistically high *in situ* production of CO.

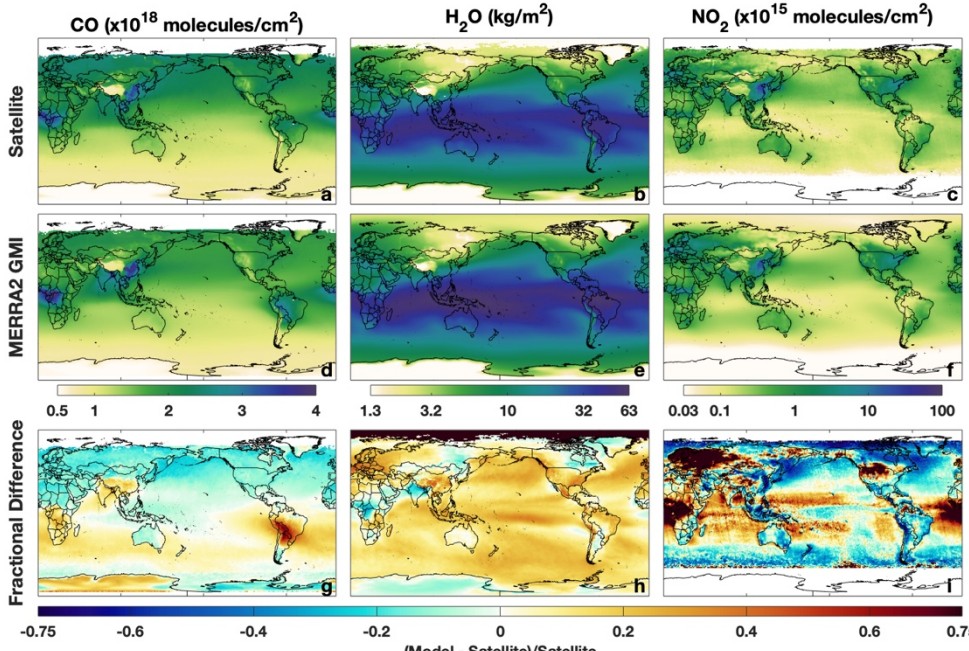

*Figure 2: Tropospheric column CO (left), $H_2O_{(v)}$ (middle), and $NO_2$ (right) from MOPITT, AIRS, and OMI, respectively (top row), and*
*MERRA2 GMI (middle row)  for DJF.  For the satellite retrievals and model, data are averaged over the time range described in the*
*text for each instrument.  The fractional difference between MERRA2 GMI and the satellite is shown in the bottom row.*
MERRA2 GMI captures the spatial distribution of $H_2O_{(v)}$, although the model is biased high in both
the column and throughout much of the troposphere.  Overestimates in column $H_2O_{(v)}$ are ~14% in
both December – February (DJF) (Fig. 2h) and JJA (Fig. 3).  These overestimates extend over most of
the world's oceans, and only small regions over northern India, central Africa, eastern Russia, and
eastern Canada show any underestimate in $H_2O_{(v)}$.  Fractional differences in $H_2O_{(v)}$ between
MERRA2 GMI and the different AIRS pressure levels are most pronounced in the tropical UT (Fig.
4).  At pressures greater than 700 hPa, modeled $H_2O_{(v)}$ is generally within 10% of the observations,
while for pressures less than 500 hPa, modeled $H_2O_{(v)}$ in the equatorial region disagrees with
observations by 55% on average.
Agreement between observed and modeled $NO_2$ is weaker than for the other species examined
here.  While MERRA2 GMI appears to capture the regions with local $NO_2$ maxima – notably those
over central Africa, eastern China, and the northeastern United States – the magnitudes frequently
differ.  The simulation shows a significant high bias over central Africa and the equatorial Atlantic
on the order of 100%, suggesting that biomass burning emissions of $NO_X$, the dominant NO source
in this region, are too high.  In contrast, concentrations over eastern Asia are too low in the model,
suggesting errors in the anthropogenic emissions inventory and/or in the $NO_X$ lifetime.  Strode et al.
(2019) also evaluated $NO_2$ in MERRA2 GMI, comparing trends in tropospheric column $NO_2$ over the
eastern US and eastern China in MERRA2 GMI and OMI.  They found that although trends were
similar between the simulation and observations in both regions, the magnitude of the trends
differed, likely due to errors in the MACCity emissions inventory.

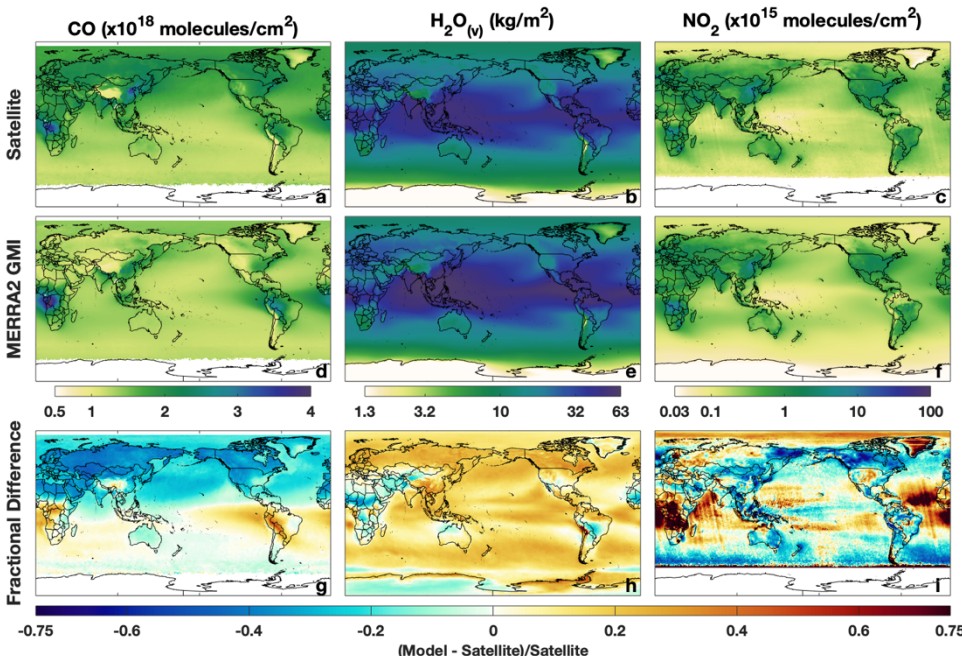

Figure 3: Same as Figure 2 except for JJA.

As with the *in situ* observations, comparison between MERRA2 GMI and satellite retrievals demonstrates that the simulation is able to capture the distribution of the chemical drivers of OH in remote regions, which tend to exhibit the strongest relationship between OH and climate modes (see Section 4.0). These results lend confidence to the analysis described in Sections 4.0 and 5.0 and suggest the findings in remote regions are likely applicable to the actual atmosphere. The large disagreement between the simulation and observed column CO and $NO_2$ in regions that are significantly impacted by biomass burning and/or biogenic emissions suggests, however, that modeled relationships of chemical species with modes of climate variability in these regions should be viewed with caution. We further evaluate the ability of the simulation to capture the relationship between ENSO and CO, $H_2O_{(v)}$, and $NO_2$ using satellite observations in Section 5.1.2.

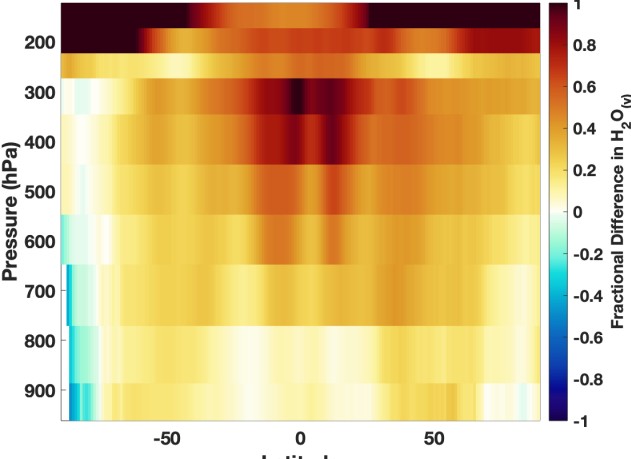

Figure 4: The fractional difference in zonal mean $H_2O_{(v)}$ between MERRA2 GMI and AIRS for the different AIRS pressure layers for DJF. Positive numbers indicate a high bias in the model.

## 4.0     The Relationship between Simulated OH Variability and Climate Modes

When considered in concert, the modes of climate variability evaluated here (i.e., ENSO, the IOD,
and NH modes) along with monsoons explain a substantial fraction of the simulated tropospheric
OH interannual variability over 19 – 40% of the global troposphere by mass, depending on season.
Figure 5 highlights regions that show significant correlation between TCOH and the NH modes
(purple), monsoons (light blue), ENSO (green), and the IOD (orange) for each season in MERRA2
GMI output.  In all seasons, correlation with ENSO has the largest spatial extent, but in DJF and
MAM, for example, the 8 NH modes can explain TCOH variability over large swaths of the NH,
comprising 10% of global, tropospheric mass.  In JJA, the combination of the different climate
modes and monsoons has the smallest spatial coverage (19% of the global, tropospheric mass),
while the IOD, consistent with its seasonal variability, only has a widespread correlation with TCOH
during SON.  Similar patterns are found for the individual layers (Fig. S2).
Below, we examine the relationships between tropospheric OH and the various modes of climate
variability demonstrated in Figure 5.  First, in Section 5.0, we show that El Niño events lead to
global reductions in tropospheric OH, with changes being driven by decreased secondary
production in the UFT that more than compensates for increased primary production in the PBL.  In
Section 6.0, we demonstrate that the effects on OH from NH modes of variability, the IOD, and some
monsoons have limited spatial scales, as compared to ENSO, but can significantly alter local OH
distributions.  In both sections, we also compare simulations from MERRA2 GMI to simulations
from the CCMI, demonstrating that the relationship between OH and climate modes is robust
among multiple models.

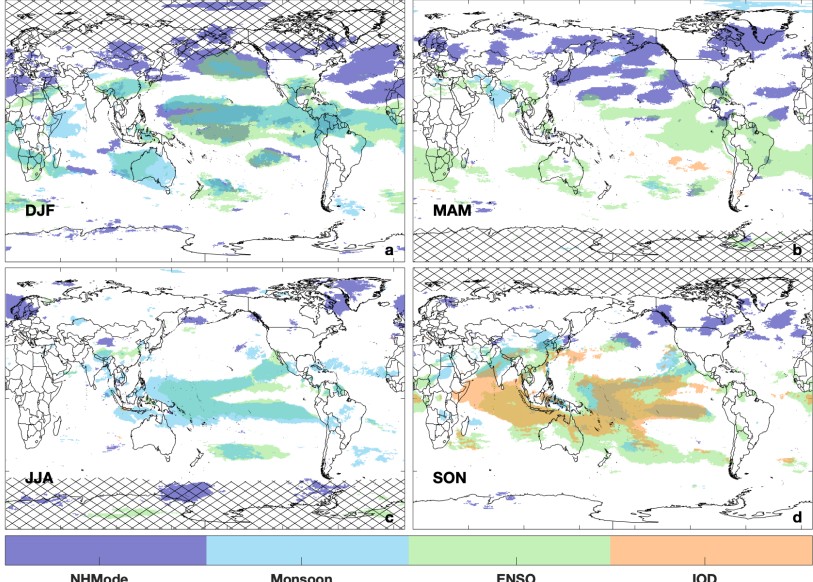

*Figure 5: Regions that show a significant correlation (absolute value of r >0.5) between a NH mode (purple), monsoon (light*
*blue), ENSO (green), or IOD (orange) and TCOH for each season in the MERRA2 GMI simulation.  Regions with TCOH less than 1 x*
*$10^{11}$ molecules/cm$^2$ have been hatched out.*

## 5.0 Relationship between Simulated OH Variability and ENSO in MERRA2 GMI
To understand the relationship between OH, its drivers, and ENSO, we first investigate the OH
production rate.  In the MERRA2 GMI simulation, the OH production rate is primarily dependent on
reactions 1 – 4, where $O^1D$ is produced from the photolysis of tropospheric $O_3$.  In the free
troposphere, these four reactions comprise at least 95% of OH production in the tropics, on
average, and at least 90% in the PBL.  Only in the regions with large biogenic emissions (e.g., South
America and central Africa) do other reactions contribute more than 15% of the total OH
production in the PBL.  As will be shown, the effects of ENSO on OH are primarily focused away
from these regions, so we restrict our analysis to reactions 1-4.

$$H_2O_2 + h\upsilon \rightarrow 2OH \qquad \text{(Reaction 1)}$$
$$NO + HO_2 \rightarrow NO_2 + OH \qquad \text{(Reaction 2)}$$
$$O_3 + HO_2 \rightarrow 2O_2 + OH \qquad \text{(Reaction 3)}$$
$$H_2O + O^1D \rightarrow 2OH \qquad \text{(Reaction 4)}$$

During El Niño events, the dominance of these individual reactions in producing OH varies with
altitude.  We focus our analysis on DJF throughout Section 5.0 because that is the season with the
largest impact of ENSO on OH as shown in Figure 5.  Figure 6 shows the zonal mean of the fraction
of total OH production from the $H_2O + O^1D$ (a) and $NO + HO_2$ (b) reactions as well as the total OH
production rate (c) during El Niño events in DJF.  While the production rates along these pathways
vary with the ENSO phase, as discussed in Sections 5.2 and 5.3, the relative importance of the
individual reactions is similar during neutral and La Niña events (not shown) and is in agreement
with previous model studies (e.g. Spivakovsky et al., 2000).
The $H_2O + O^1D$ reaction is dominant from the surface to about 800 hPa through much of the SH and
the tropics, while, near the surface, the $NO + HO_2$ reaction only has large impacts in the NH mid-
latitudes.  This influence of $NO_X$ in the NH mid-latitudes extends through much of the troposphere.
In the UFT, this reaction is the greatest contributor to total OH production at all latitudes except the
NH polar region, where the $HO_2 + O_3$ reaction dominates during polar night (Fig. S3).  Total OH
production in the polar regions, however, is orders of magnitude lower than in the tropics.  Outside
of the polar regions, the $HO_2 + O_3$ and $H_2O_2$ photolysis reactions generally contribute between 10
and 30% of the total rate (Fig. S3).  The dominant OH sink throughout the troposphere is CO, which
is responsible for 50% or greater of OH loss at all tropospheric pressures and latitudes (Fig. S4)
during El Niño events.  Because of the differing importance of the individual OH production
reactions with altitude, we first examine the relationship between OH and ENSO for TCOH (Section
5.1) and then separately for the PBL (Section 5.2) and the UFT (Section 5.3).   Finally, in Section 5.4,
we investigate the MFT and LFT, where the effects of ENSO on OH are more limited.

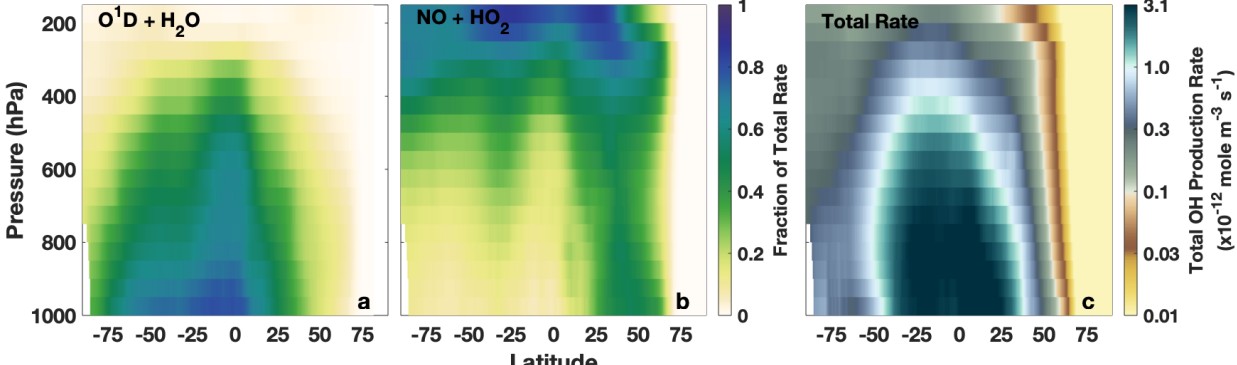

*Figure 6: Zonal mean of the fractional contribution of the $O^1D + H_2O$ (a) and  $NO + HO_2$ (b) reactions to the total OH production*
*rate as well as the total OH production rate (c) for El Niño events (MEI > 0.5) for DJF averaged over 1980-2018.*
**5.1    Tropospheric Column OH**
**5.1.1    The Relationship between Simulated TCOH and ENSO**

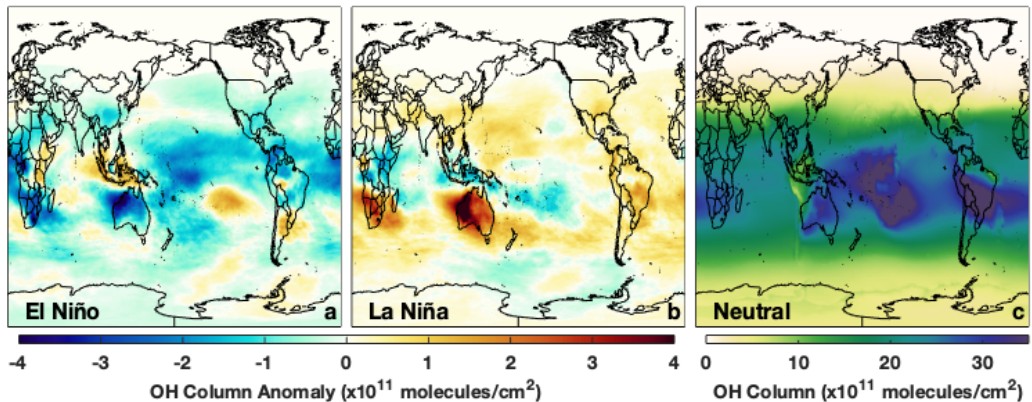

*Figure 7: Absolute difference in TCOH between El Niño events and neutral events (a) for DJF averaged over 1980 – 2018. El Niño and neutral events are defined as a season having an MEI value greater than 0.5 or an MEI value between -0.5 and 0.5, respectively. The analogous plot for La Niña events (MEI less than -0.5) is also shown (b). Panel c shows the average OH column for neutral events. The 1980 – 2018 time period includes 11 El Niños, 12 La Niñas, and 15 neutral events in DJF.*

As shown in Figure 7, TCOH decreases by 3.3% during El Niño events (relative to neutral events) equatorward of 30° in DJF and is characterized by widespread decreases in the tropics and subtropics, especially northern Australia, and west-central and southern Africa. Regional increases are found over eastern Africa, the east-central Pacific, southern South America, and Indonesia. Maximum decreases in TCOH are on the order of $4.5 \times 10^{11}$ molecules/cm$^2$ ($\sim$10-15%) and are centered over northern Australia, while maximum increases in TCOH ($\sim 2.5 \times 10^{11}$ molecules/cm$^2$) are centered over Sumatra.

During La Niña events, TCOH increases relative to neutral events over much of the globe, although the changes are not necessarily symmetric with those seen during El Niño events. Increases over Australia are on the order of 1 to $2 \times 10^{11}$ molecules/cm$^2$, on par with the decreases seen during El Niño, but the changes during La Niña are centered over Western Australia and the Indian Ocean. Over the Pacific, the magnitude of the OH increase is lower (on the order of 0.5 to $1 \times 10^{11}$ molecules/cm$^2$) than the decreases found during El Niño, and some regions off the coast of Hawaii and Papua New Guinea show decreases during both ENSO phases. Besides these two regions, there are also significant decreases in OH over eastern Africa and in the southern portion of South America.

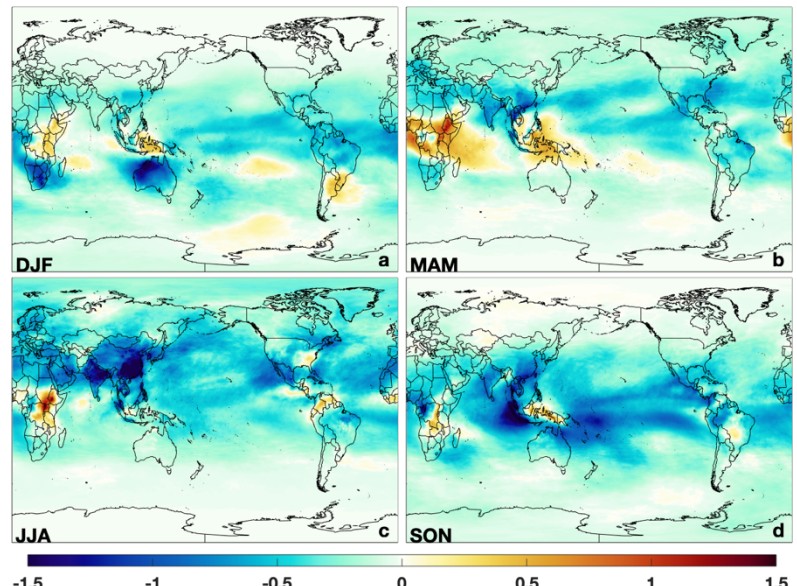

*Figure 8: The first EOF of TCOH from MERRA2 GMI for DJF (a), MAM (b), JJA (c), and SON (d).*
Consistent with these widespread changes in TCOH, EOF analysis demonstrates that over most
seasons, JJA being the notable exception, ENSO is the dominant mode of OH variability. Figure 8
shows the spatial component of the first EOF of TCOH for the four seasons. While EOF analysis does
not quantify changes in column content, it does highlight, for each mode of variability, regions
where changes in TCOH are most prominent. For DJF, the first EOF (Fig. 8a) is almost identical to
the composite figure showing OH anomalies during El Niño (Fig. 7a). Likewise, the temporal
component of the 1st EOF strongly correlates with the MEI ($r^2$ = 0.70, Table 2). In DJF, the first EOF
is responsible for 29% of the total spatial variance for TCOH. Although ENSO is the dominant mode,
however, 70% of the spatial variance is still unexplained. In JJA, ENSO influence on OH is much
weaker, with a correlation between the 1st EOF and TCOH of $r^2$ = 0.25, consistent with the seasonal
cycle of ENSO.
While the spatial pattern of the EOF varies seasonally (Fig. 8), ENSO shows similar levels of
correlation to the temporal component of the 1st EOF in MAM and SON as for DJF, with $r^2$ values of
0.54 and 0.60, respectively. Likewise, the spatial patterns of the first EOF of TCOH for these seasons
are similar to the composite figures showing OH anomalies during El Niño (Fig. S5). For MAM,
again the EOF shows regions with a negative sign over much of the Northern Hemisphere, with the
largest magnitude centered over the Pacific Ocean, India, and Atlantic coast of the United States.
Regions with an opposite sign include the Maritime Continent and much of central Africa. In SON,
almost all of the tropics show some response, with major centers off the east coast of Papua New
Guinea and off the west coast of Sumatra. In addition, there is a larger response over the Indian
Ocean than for other months, also evident in the regression of TCOH with the MEI, suggesting the
possible influence of the IOD, which is correlated with ENSO ($r^2$ = 0.30). This seasonal component
in the strength of the relationship between the EOF and the MEI is also reflected in the correlation
analysis (Fig. 5), where the area of correlation between TCOH and the MEI maximizes in DJF and
minimizes in JJA.
*Table 2: For each season, we show the $r^2$ of the correlation of the temporal component of the EOF that has the highest*
*correlation with the MEI for TCOH and for OH in each layer. In addition, we also indicate the percent of the total spatial*

*variance explained by that EOF. With the exception of the values indicated by a \*, the 1st EOF has the highest correlation with the MEI. Those indicated with a \* are the 2nd EOF.*

| Month | Column | | UFT | | MFT | | LFT | | PBL | |
|---|---|---|---|---|---|---|---|---|---|---|
| | *Pct. Variance* | *$r^2$* | *Pct. Variance* | *$r^2$* | *Pct. Variance* | *$r^2$* | *Pct. Variance* | *$r^2$* | *Pct. Variance* | *$r^2$* |
| DJF | 29.4 | 0.7 | 37.6 | 0.73 | 20.8 | 0.81 | 11.7* | 0.55 | 12* | 0.85 |
| MAM | 25.9 | 0.54 | 36.2 | 0.61 | 23.4 | 0.40 | 9.5* | 0.48 | 9.3* | 0.59 |
| JJA | 30.7 | 0.25 | 44.6 | 0.14 | 29 | 0.15 | 27.7 | 0.06 | 39.4 | 0.07 |
| SON | 33.2 | 0.60 | 41.1 | 0.50 | 22.8 | 0.63 | 12.3* | 0.59 | 9.3* | 0.63 |

## 5.1.2 The Relationship between TCOH Drivers and ENSO

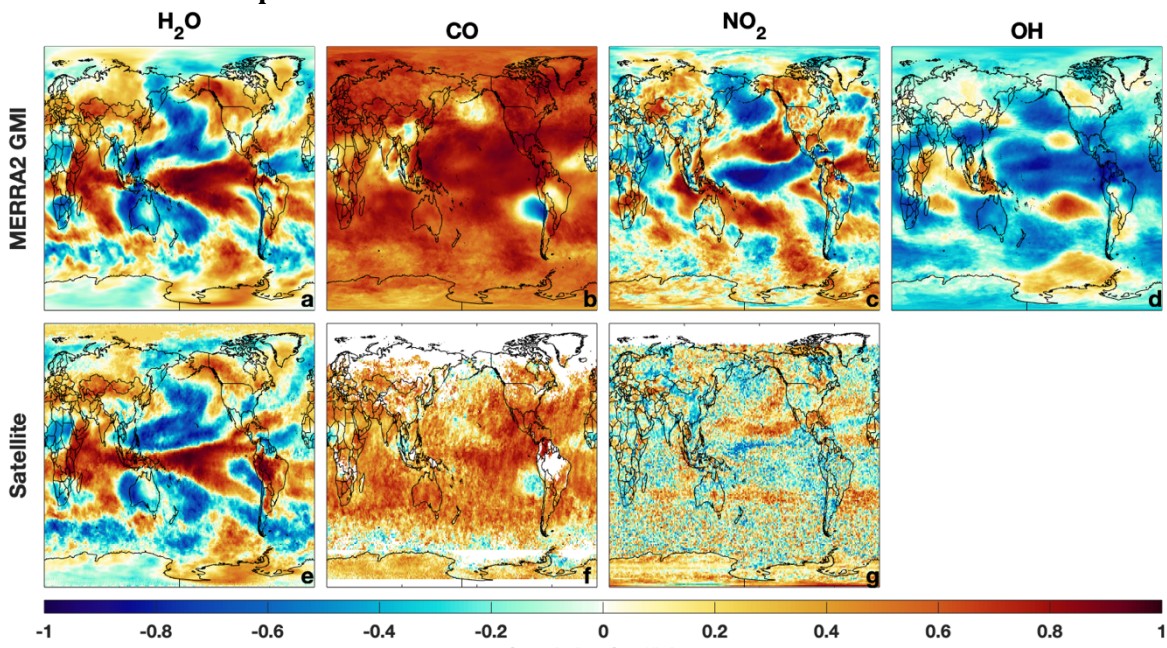

*Figure 9: Regression of tropospheric column $H_2O_{(v)}$ (a), CO (b), $NO_2$ (c), and OH (d) from MERRA2 GMI (top) and satellite retrievals from AIRS (e), MOPITT (f), and OMI (g) against the MEI for DJF over the satellite lifetime.*

To understand the factors driving ENSO-related changes in TCOH, we also investigate the relationship between OH precursors and ENSO. Figure 6 demonstrates that the $O^1D + H_2O$ and $NO + HO_2$ reactions control zonal mean OH production in the tropics. As a result, we investigate the relationship between tropospheric column $H_2O_{(v)}$, CO, $NO_2$ and ENSO using both MERRA2 GMI output and satellite retrievals. We use $NO_2$ here, instead of NO, because of its observability from space, although simulated NO demonstrates similar spatial correlation patterns with the MEI as simulated $NO_2$.

Regression of total column $H_2O_{(v)}$ from AIRS against the MEI (Fig. 9e) reveals a tri-pole pattern over the Pacific Ocean, with an area of positive correlation throughout much of the equatorial Pacific Ocean and areas of anti-correlation poleward of this region, in agreement with previous work (e.g. Shi et al., 2018). Each of these areas is well-captured by the MERRA2 GMI simulation (Fig. 9a), showing nearly identical spatial patterns and strength of correlation over most of the globe. This relationship between $H_2O_{(v)}$ and ENSO can be explained by the increased convective uplifting in the equatorial Pacific and associated increased subsidence poleward of this region during El Niño events. While the anticorrelation between $H_2O_{(v)}$ and the MEI over Australia and southern Africa is

consistent with the decrease in TCOH over these regions during El Niño events (Fig. 7), the positive
correlation between $H_2O_{(v)}$ and the MEI over the equatorial Pacific suggests there must be
competing effects from other OH drivers in order to explain the decreases in TCOH in this region.
Simulated tropospheric column $NO_2$ is strongly anti-correlated with ENSO over the equatorial
Pacific, indicating a suppression of OH production when the MEI is positive (El Niño), consistent
with Figure 7. Column $NO_2$ exhibits the opposite correlation pattern as $H_2O_{(v)}$ over the Pacific, with
decreases in $NO_2$ in regions with increased $H_2O_{(v)}$ and vice versa. The similarities in the spatial
correlation patterns for $NO_2$ and $H_2O_{(v)}$ with the MEI suggests that convection is also at least
partially driving the changes in $NO_2$ in the equatorial Pacific. Changes in the Walker Circulation
associated with El Niño events have been shown to redistribute $O_3$ in the tropics, resulting in a
dipole pattern over the western and central Pacific (Oman et al., 2011). Analysis of vertical winds
and the $NO_2$ anomaly suggests a similar mechanism for $NO_2$.
Correlations between OMI $NO_2$ and the MEI suggest similar relationships as found in the MERRA2
GMI simulation, although the correlations are not as robust as for the other satellite variables
examined here. This is likely because tropospheric $NO_2$ columns over the ocean are frequently at or
below the instrumental average noise ($5 \times 10^{14}$ molecules/cm$^2$). As with the simulation, OMI
suggests broad regions of anti-correlation between ENSO and $NO_2$ in the equatorial Pacific and Gulf
of Alaska as well as a region of positive correlation in the extra-tropical NH Pacific. These results
demonstrate that, with enough temporal and spatial averaging, OMI is capable of capturing the
variability of tropospheric $NO_2$ even in remote regions with low concentrations.
Tropospheric column CO and the MEI are positively correlated over most of the globe in both
MERRA2 GMI and in MOPITT (Figs. 9b and f, respectively), suggesting strong increases in CO during
El Niño events. This increase in CO is associated with increased biomass burning, particularly in
Indonesia, and is consistent with the modeled decrease in OH (e.g. Duncan, 2003a) and with the
widespread decrease in TCOH over much of the tropics.

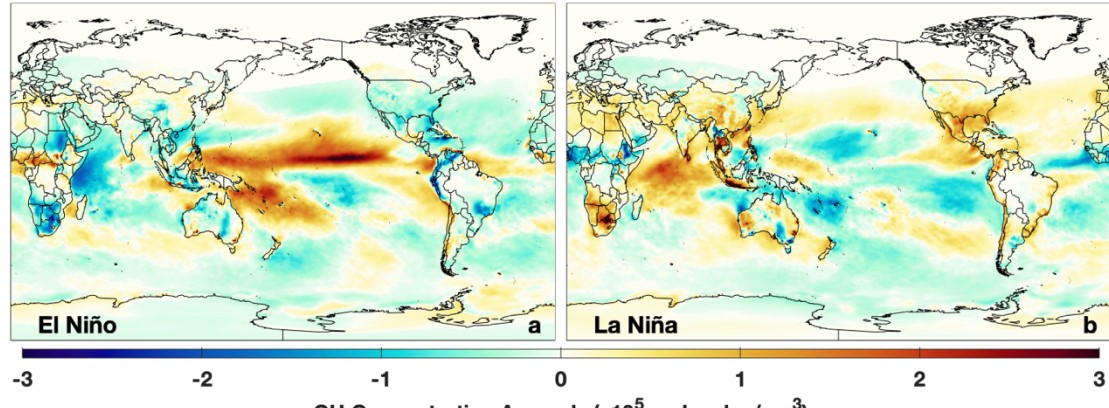

*Figure 10: Same as panels a and b of Figure 7 except for the PBL level.*
**5.2     The Planetary Boundary Layer**
**5.2.1 The Relationship between PBL OH and ENSO**
In contrast to the tropospheric column (Fig. 7), mean mass-weighted OH (e.g., Lawrence et al.,
2001) in the PBL increases globally by 1% during El Niño events (Fig. 10), although regional
differences are significantly larger. PBL OH exhibits an area of strong positive correlation with the
MEI (Fig. 11d) over the central Pacific, marked by increases in concentrations on the order of 2-3 ×
$10^5$ molecules/cm$^3$, approximately 15% higher than concentrations in neutral events. Changes in
the PBL during La Niña are smaller, with localized concentration decreases of about 5 – 10% over
the tropical Pacific (Fig. 10b).  Regions with significant correlation between PBL OH and the MEI are
distinctly smaller than in the UFT (Fig. 11) and for TCOH (Fig. 5a), further emphasizing the
comparatively limited spatial effects of ENSO in the PBL.

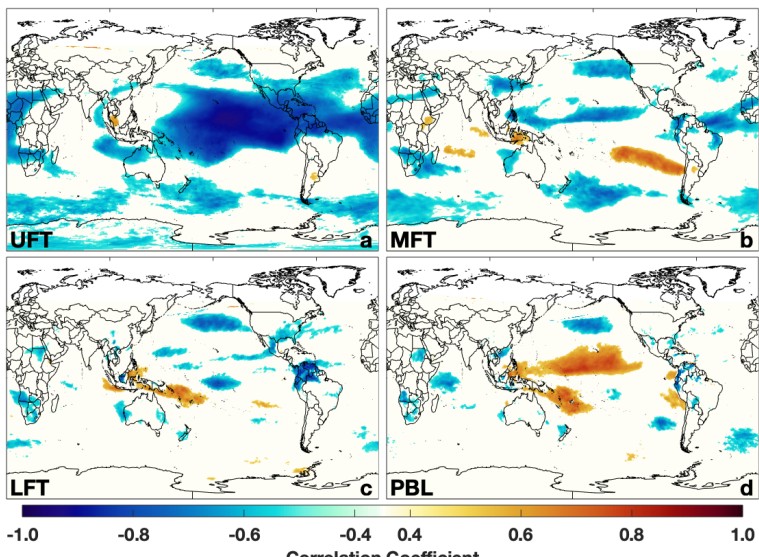

*Figure 11: Correlation of OH from MERRA2 GMI with the MEI for the different atmospheric layers in DJF.*
The more geographically limited changes in OH shown by the composite and regression analyses
are consistent with EOF analysis.  During all seasons except JJA, ENSO correlates more strongly with
the 2nd EOF for the PBL (Table 2), suggesting another mechanism is the dominant mode of
variability.  The spatial pattern of the $2^{nd}$ EOF for PBL OH varies markedly across seasons (Fig. S6),
with the largest signal over the tropical Pacific during DJF and MAM and over Indonesia in SON.  In
general, the $r^2$ with ENSO is 0.5 or higher and the mode contributes approximately 10% of the total
spatial variance, although correlation in JJA ($r^2$ = .07) is negligible.
In contrast to the ENSO-related EOFs, the first EOF (Fig. S7) for the DJF PBL layer reveals a spatial
pattern much more limited to continental regions and areas of continental outflow, suggesting that
this mode of variability is potentially reflective of long-term emission trends, in both anthropogenic
and biomass burning emissions.  This is more evident in the 1st EOF for JJA, where the spatial
pattern shows opposite signs over regions with known net emissions reductions (the United States,
portions of Europe, and Japan) and those with known net emissions increases (China, India, the
Middle East) over the 1980 – 2018 period examined here.
**5.2.2 The Relationship between PBL OH drivers and ENSO**
Approximately 80% of the zonal mean OH production in the tropical PBL during El Niño events is
from the $H_2O + O^1D$ reaction (Fig. 6a).  Figure 12 shows the correlation of the MEI against both OH
production from this reaction as well as the total OH production rate for the PBL.  Similar plots for
the other OH production reactions are shown in Figure S8.  The nearly identical regression pattern
for the $H_2O + O^1D$ and the total production rate with the MEI demonstrates that changes in this
reaction are driving changes in OH in the tropics during El Niño events.

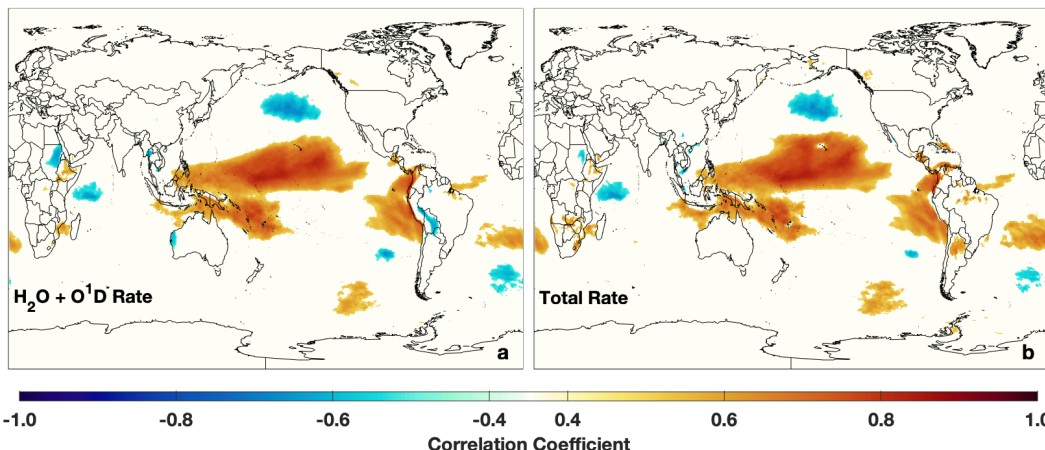

*Figure 12: Correlation of the MEI with the production rate of OH from the $H_2O + O^1D$ reaction (a) for DJF and the total OH*
*production rate as defined in the text (b) for the PBL level are shown.*
To understand the relationship between the OH production rate and ENSO in the PBL, we examine
the changes in $H_2O_{(v)}$ and $O^1D$ (Fig. 13). The spatial correlation of $H_2O_{(v)}$ and the MEI in the PBL
exhibits a tripole pattern similar to that seen in the tropospheric column (Fig. 9a). While $H_2O_{(v)}$ is
correlated with the MEI in the equatorial Pacific, which would lead to increases in OH production,
$H_2O_{(v)}$ is anti-correlated with the MEI near the Hawaiian Islands and in the south Pacific, which
would lead to decreased OH production in these regions. Because OH increases in these areas
during El Niño events, the decreased $H_2O_{(v)}$ is offset by increases in $O^1D$ to result in a net positive
correlation of the total OH production rate.

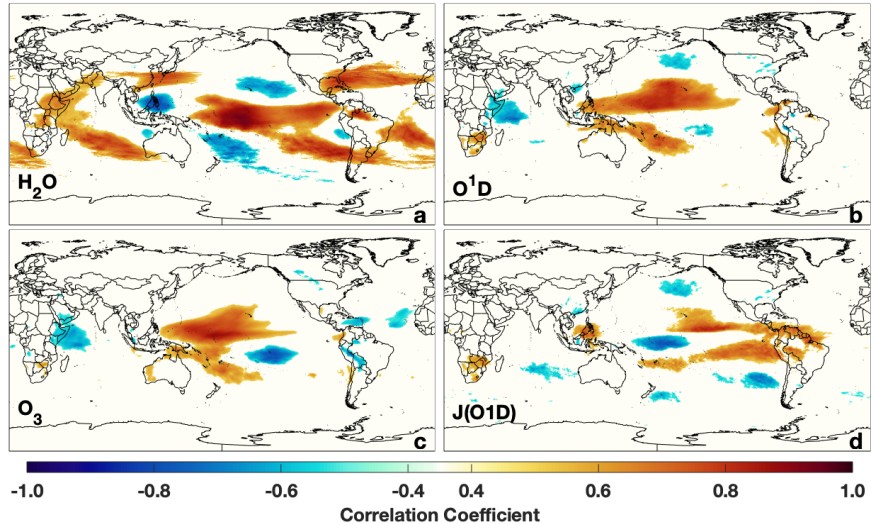

*Figure 13: Correlation of the indicated species with the MEI for the PBL level for DJF.*
Changes in $O^1D$ and its photochemical drivers, $O_3$ and the rate of $O_3$ photolysis to $O^1D$ ($J(O^1D)$), are
driving the ENSO-related changes in OH in the PBL. $O^1D$ shows distinct regions of positive
correlation with ENSO extending from the Philippines to the eastern Pacific Ocean and another
region of positive correlation off the coast of Papua New Guinea (Fig. 13b). $O^1D$ abundance is
controlled both by $O_3$ concentrations and incoming solar radiation at wavelengths less than 320
nm. Positive correlation between ENSO and $O_3$ in the PBL is limited to the western Pacific Ocean,
where horizontal advection of relatively high $O_3$ air from Indonesia to the Pacific Ocean is increased
during El Niño events due to changes in the Walker Circulation (Oman et al., 2011). Changes in $O_3$
and O$^1$D off the coast of Papua New Guinea are potentially linked to the South Pacific Convergence
Zone, which has a strong dependence on ENSO (Borlace et al., 2014).  J(O$^1$D) exhibits two regions of
positive correlation extending from South America, one that reaches Hawaii in the NH and another
that spans almost to the coast of Australia in the SH (Fig. 13d).  The MERRA2 GMI simulation shows
reduction in total stratospheric column O$_3$ of 2-5% in the tropics during El Niño, consistent with
previous work  (e.g., Randel et al., 2009), which could contribute to the increase in J(O$^1$D), although
more work is needed to establish this link.
### 5.3     The Upper Free Troposphere
#### 5.3.1   The Relationship between UFT OH and ENSO

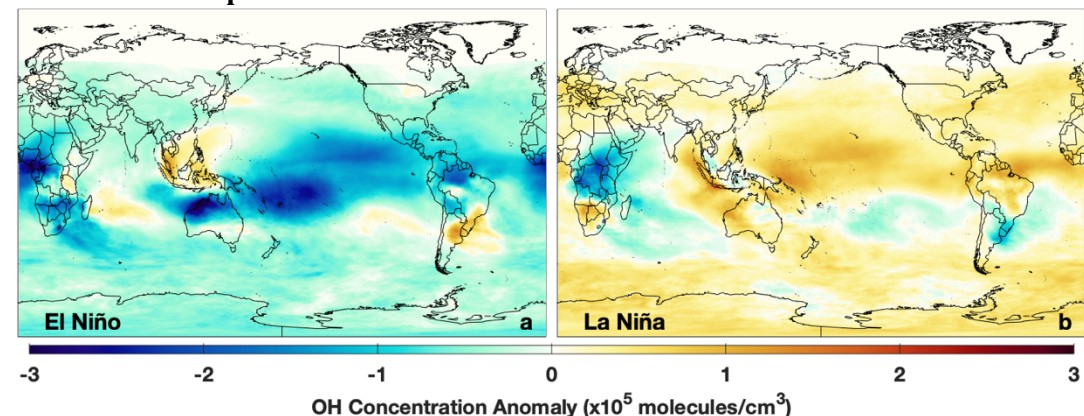

OH Concentration Anomaly (x10$^5$ molecules/cm$^3$)

*Figure 14: Same as Figure 7 except for the UFT.*
Similar to the relationship between ENSO and TCOH, OH in the UFT shows a strong anticorrelation
with the MEI over much of the tropics (Fig. 11a) resulting in large-scale decreases during El Niño
events.  Decreases are highest over Northern Australia and the west-central Pacific, on the order of
1-2 × 10$^5$ molecules/cm$^3$ or 15-20% lower than in neutral events.  During La Niña events, OH
increases with respect to neutral events over much of the globe, although the magnitude of the
increases is lower than for El Niño events.  As with TCOH, one notable exception is over central
Africa, where UFT OH decreases between 1-2 × 10$^5$ molecules/cm$^3$.
EOF analysis on UFT OH followed by correlation of the temporal component (*i.e.,* the principal
component) with the MEI demonstrates that ENSO is the dominant mode of OH variability in the
UFT throughout much of the year.  The MEI correlates with UFT OH ($r^2 > 0.5$) for DJF, MAM, and
SON, and explains 36% of the spatial variance or greater in each of the seasons (Table 1),
demonstrating that the relationship between ENSO and OH is even stronger in the UFT than in the
tropospheric column as a whole.  As with the other atmospheric levels, there is little correlation
between OH and the MEI for JJA.
#### 5.3.2 The Relationship between UFT OH drivers and ENSO
While changes in the O$^1$D + H$_2$O reaction drive ENSO-related changes in OH production in the PBL,
the NO + HO$_2$ reaction drives OH production in the UFT.  The nearly identical correlation patterns
between the NO + HO$_2$ reaction (Fig. 15) and the total OH production rate in the UFT layer suggest
that changes in NO and/or HO$_2$ during El Niño are driving interannual OH variability in the UFT,
leading to decreased OH production over most of the tropical Pacific.  This dependence on the NO +
HO$_2$ reaction is consistent with its overall contribution to the total production rate as shown in
Figure 6.  Similar plots for the other OH production reactions are shown in Figure S9.   While J(O$^1$D)
does increase in the UFT during El Niño events, as does production from the O$^1$D + H$_2$O reaction in
some regions, the relatively small contribution of this reaction to the total OH production in the
UFT (Fig. 6a) does not significantly perturb OH in this layer.
Regression analysis suggests that changes in NO are driving the relationship between OH and ENSO
in the UFT in MERRA2 GMI.  The MEI-NO correlation exhibits a strong dipole pattern in the tropics
(Fig. 16), with areas of positive correlation over southeast Asia and the maritime continent and a
large area of anti-correlation over much of the Pacific.  $HO_2$ exhibits the opposite pattern, with
increased concentrations over much of the Pacific during El Niño.  This is consistent with the NO
pattern, as decreased NO concentrations favor partitioning of $HO_X$ ($HO_X = OH + HO_2$) towards $HO_2$.

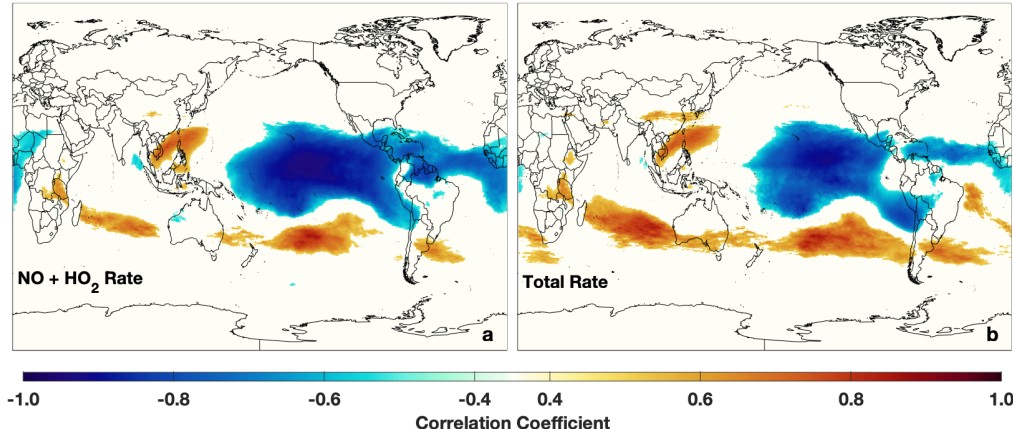

*Figure 15: Correlation of the production rate of OH from the NO + $HO_2$ reaction (a) for DJF and the total OH production rate as*
*defined in the text (b) with the MEI for the UFT level are shown.*
Similarities between NO and $O_3$ correlation with the MEI in the UFT suggest similar mechanisms in
controlling the spatial distribution of these species.  The relationship between $O_3$ and the MEI
shown in Figure 16b is similar to that found in Oman et al. (2013) using satellite data.   They
demonstrated that areas of increased $O_3$ over Indonesia coincided with increased downward flow
in the region associated with changes in the Walker circulation.  Decreases in $O_3$ over the Pacific
coincided with increased upward motion, convectively lofting low $O_3$ air throughout the column.
Similarly, regions of anomalously low NO in the UFT during El Niño events are associated with
regions of anomalous upward motion, suggesting that decreases in upper tropospheric NO results
from the convective lofting of $NO_X$-poor air from lower in the tropospheric column.

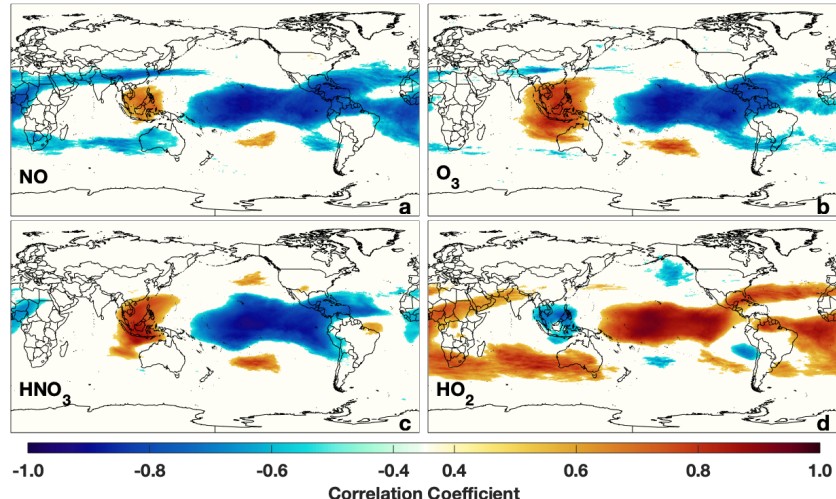

Figure 16: Correlation of the indicated species with the MEI for the UFT level for DJF.

The anti-correlation between ENSO and NO also suggests that lightning emissions of NO over the tropical Pacific do not significantly increase OH production in the region during El Niño events. Lightning NO emissions in MERRA2 GMI show a correlation pattern (Fig. 17) similar to that of $H_2O_{(v)}$ (Fig. 9a), with increased lightning over the equatorial Pacific and decreased lightning poleward of this region during El Niño events. The correlation pattern from MERRA2 GMI output agrees closely with flash rate data observed from the Lightning Imaging Sensor (LIS). The only region of significant difference between the satellite and MERRA2 GMI is in the equatorial Pacific, where the region of positive correlation extends from Papua New Guinea to the South American coast in the simulation but only about half that distance in the satellite product.

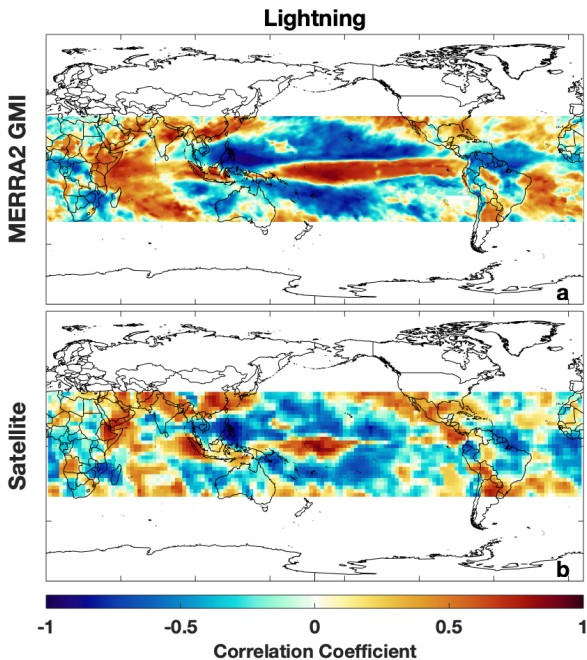

Figure 17: The regression of lightning NO emissions at 300 hPa (a) and the lightning flash rate from the LIS/OTD time series (b) against the MEI. Lightning data are restricted to within 35 degrees of the equator because of the spatial coverage of the Tropical Rainfall Monitoring Mission (TRMM) satellite, on which LIS is located.

This tri-pole correlation pattern between MEI and lightning, evident in both the satellite and model
(Fig. 17) is in contrast to the relationship with NO (Fig. 16a) and other reactive nitrogen ($NO_y$)
species in the UFT.  While the anti-correlation in NO is consistent with the changes in lightning NO
emissions in some regions, in the equatorial Pacific band, NO decreases during El Niño events
despite an increase in lightning NO emissions.  This apparent discrepancy occurs because even
though lightning NO increases by 100% or more over the equatorial Pacific during El Niño events in
the model, the absolute difference is orders of magnitude lower than the accompanying changes
over land.  We conclude that the resulting NO perturbations over the equatorial Pacific latitudes are
dominated by mechanism other than the local lightning response, such as changes in the Walker
Circulation and the associated transport of air originating over the continents.  This mechanism is
supported by the similar regression pattern of longer-lived species, such as $HNO_3$ (Fig. 16c) and
PAN (not shown), to NO in the UFT, showing that transport of reactive nitrogen from other source
regions, particularly lightning over South America, is likely reduced during El Niño events.
Our findings are broadly consistent with Turner et al. (2018), who found that increases in lightning
NO emissions drive increases in OH during La Niña and, conversely, decreases in lightning NO
emissions lead to OH decreases during El Niño.  The results presented here suggest that in addition
to this influence of lightning locally, other mechanisms, such as atmospheric transport of $NO_y$
species, also likely contribute to the relationship between ENSO and OH in the equatorial Pacific.
**5.4     Variability in the MFT and LFT**
As in the UFT, ENSO is the dominant mode of variability in the MFT in DJF, with strong correlation
between the MEI and the temporal component of the first EOF ($r^2 = 0.81$) and the first EOF
explaining 20.8% of the total spatial variance.  Likewise, the largest OH anomalies in the LFT during
both El Niño and La Niña are centered over Australia and South Africa (Fig. 18), similar to patterns
seen in the UFT.  Unlike in the UFT, however, there is a large region extending from the coast of
South America into the Pacific where OH concentration is positively correlated with ENSO.  These
changes are driven by increase in $H_2O_{(v)}$, and subsequent increased OH production from the $H_2O$ +
$O^1D$ reaction.

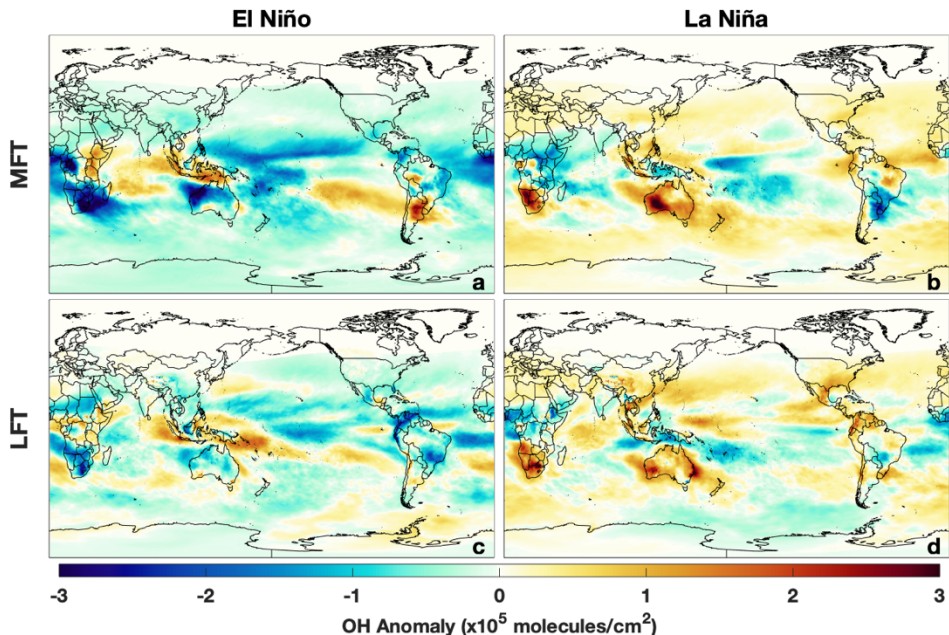

*Figure 18: Same as panels a and b of Figure 7 except for the MFT and LFT.*
ENSO-related changes in OH concentration in the LFT are smaller in magnitude than for the other
atmospheric levels (Fig. 18), with maximum increases in OH during El Niño on the order of 1 – 1.5 x
$10^5$ molecules/cm$^3$. The spatial extent of significant correlation between the MEI and OH
concentration in the LFT is smaller than for the other atmospheric levels (Fig. 11), with the most
prominent feature being an area of positive correlation near Indonesia. Consistent with the more
limited impact, ENSO is correlated with the 2nd EOF of OH concentration for the LFT ($r^2$ = 0.55),
explaining only 11.7% of the total variability (Table 1).
It is likely that competing effects from the different drivers limit the interannual variability in OH in
the LFT and MFT, explaining the smaller regions of correlation with ENSO. For these levels, no
single OH production reaction clearly explains the relationship between ENSO and OH. In contrast
to the PBL and UFT, where the relationship between the total OH production rate closely mirrored
the production rates from the $O^1D + H_2O$ and $NO + HO_2$ reactions, respectively, there are no
analogous relationships for the LFT and MFT. At these levels, no reaction clearly dominates total
OH production (Fig. 6). Increases in $H_2O$ in the mid troposphere, which would tend to increase OH,
are offset by decreases in NO and $O_3$. These competing effects likely explain why the absolute
changes in OH are comparatively smaller in the LFT than in the other layers.
The comparatively smaller changes in LFT OH during El Niño events limit the effect of ENSO on the
interannual variability of the $CH_4$ lifetime. Global mean, mass-weighted tropospheric OH decreases
by 2.2% during El Niño events, corresponding to only a 1% decrease in the $CH_4$ lifetime. While
changes in OH concentration are most pronounced in the UFT and PBL, $CH_4$ lifetime is mostly
dictated by OH in the LFT due to the temperature dependence of the $OH + CH_4$ reaction rate. This
limited effect on $CH_4$ lifetime highlights the importance of investigating the spatial OH variability as
global mean metrics can obscure important year-to-year changes.
**5.5    Comparing Simulated OH Relationships with ENSO in MERRA2 GMI with the CCMI**
**        models**
To understand whether the relationship between OH and ENSO found in MERRA2 GMI is robust, we
examine model simulations from the CCMI. To compare the relationship between OH and ENSO
among the different models, we performed the same regression analysis on TCOH for the 4 CCMI
models considered here as for MERRA2 GMI. Figure 19 shows the number of CCMI models that
demonstrate a meaningful correlation between TCOH and the MEI, defined as the absolute value of
r greater than 0.5, for each grid cell. To facilitate comparison, OH for each model has been
regridded to the resolution of the model with the lowest horizontal resolution (2.81° longitude x
2.77° latitude). This regridding does not substantially alter the correlation patterns examined here.

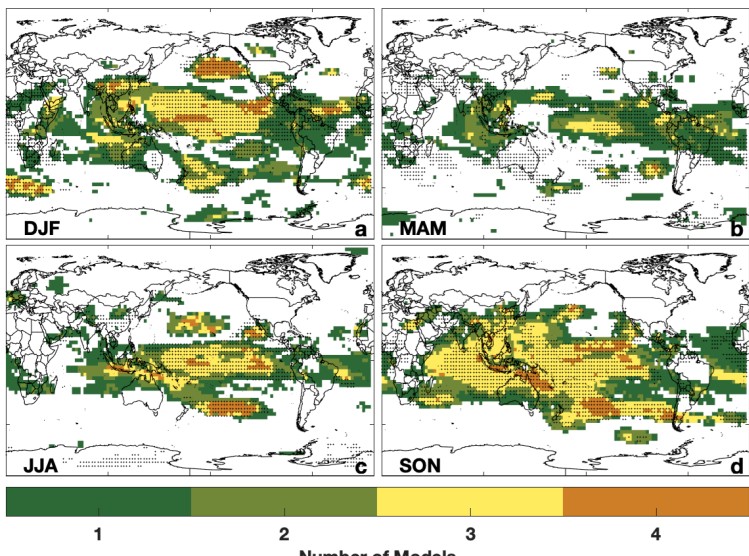

*Figure 19: The number of CCMI models that show a correlation between TCOH and ENSO over the period 1980 to 2010. Only*
*regressions with an absolute value of r greater than 0.5 are included. All models have been regridded to a common horizontal*
*grid. This regridding does not substantially alter the correlation patterns examined here. Grid boxes that also exhibit significant*
*correlations between TCOH and ENSO for MERRA2 GMI are indicated by the stippling.*
In agreement with MERRA2 GMI, TCOH varies with ENSO over a large fraction of the tropics in most
of the CCMI models, with broadly similar spatial regression patterns for most models across all
seasons except for MAM (Fig. 19). In DJF, most models show strong correlation between ENSO and
column OH over the central Pacific and south of the Aleutian Islands, with at least three CCMI
models and MERRA2 GMI showing correlation in each of these areas. This agreement highlights the
relationship of OH with ENSO as well as with the PNA and Australian monsoon, as discussed in
Section 6.0. Similar agreement among models was found for SON and JJA, although the spatial
extent of the highly correlated region is much smaller for JJA. In SON, the expansion of the area of
significant correlation over most of the Indian Ocean likely results from the strong relationship
between the IOD and ENSO during this season. There is less agreement in MAM, with only 1 or 2
models showing strong correlation in most regions.
EOF analysis of the different CCMI models likewise suggests that, in DJF, ENSO is the dominant
mode of TCOH variability. The spatial pattern of the first EOF of TCOH in DJF for the five models is
shown in Figure S10, and the principal component time series, along with the time series of the
MEI, is shown in Figure S11. MERRA2 GMI, WACCM, and MRI show a strong correlation between
the MEI and the first EOF ($r^2 > 0.64$). For each of these models ENSO is the cause of 29 – 48% of the
total spatial variance in TCOH. The correlation between the first EOF and the MEI for CHASER is
weaker ($r^2 = 0.28$), although the spatial component shows similarities to the other models.
Correlation between the MEI and the EOFs for the UFT and MFT levels increases to 0.56 and 0.45,
respectively, showing that ENSO is still important in controlling the interannual variability of
CHASER, at least in the UFT. Similarly, EMAC has no correlation between the 1st EOF of TCOH and
the MEI, but does for the UFT layer ($r^2 = 0.64$). This EOF explains 20% of the total spatial variance
for this level but has a substantially different spatial pattern than for the other models. While
further work is needed to understand the cause of the relationship between OH and ENSO in the
UFT in EMAC, results from MERRA2 GMI suggest a role for changes in production via the $NO + HO_2$
reaction.

The agreement among the majority of the models suggests that the relationship between ENSO and
TCOH is robust.  While SSTs and emissions are identical among the models, meteorology, chemical
mechanisms, and parameterizations, such as that for lightning and convection, vary widely.  Despite
the differences in these chemical and dynamical drivers of OH, the spatial patterns of the ENSO
TCOH relationship are similar for most models.  While it is beyond the scope of this paper,
determining the cause of inter-model differences in this relationship between OH and climate
modes could further our understanding of the mechanisms driving interannual OH variability.
Given the results from the MERRA2 GMI analysis, investigating ENSO-related changes in UFT NO,
both from lightning and transport, could provide insight into these inter-model differences.
Further, Nicely et al. (2020) showed that $J(O^1D)$ was the largest driver in differences in the methane
lifetime in the CCMI models, suggesting the potential importance of this variable in inter-model
differences in the OH-ENSO relationship in the PBL and lower troposphere.
**6.0 Relationship between simulated OH and NH Climate Modes, Monsoons, and the IOD**
We now investigate the relationship between OH and the NH modes of variability, monsoons, and
the IOD.  In Section 6.1, we evaluate the relationships in MERRA2 GMI, demonstrating that these
other climate features exert a much more spatially limited influence on OH as compared to ENSO
(Fig. 5).  Despite the comparatively limited extent of influence, each of these modes of variability
can strongly influence the atmospheric oxidative capacity on the local scale.  In Section 6.2, we
compare the results from MERRA2 GMI to CCMI simulations, demonstrating that the relationship
between OH and the IOD and NH modes is robust among models, while the relationship between
monsoons and OH is primarily limited to MERRA2 GMI.
**6.1 Simulated OH and the NH Climate Modes, Monsoons, and the IOD in MERRA2 GMI**
Northern Hemispheric modes of variability are strongly correlated (r>0.5) with OH over ~10% of
the globe during DJF but have a comparatively smaller effect on global OH than ENSO.  During the
positive phases of the NAO, defined as the index being greater than 0.4, TCOH increases by up to
25% in the northern Atlantic.  Similarly, during the positive phase of the PNA, TCOH decreases by
10 – 20% in the northern Pacific (Fig. 20).  Because OH production is almost an order of magnitude
lower in the NH mid-latitudes than in the tropics (Fig. 6c), however, the resultant decrease in global
mean mass-weighted OH (e.g., Lawrence et al., 2001) during the positive phase of the NAO is only
0.77%, as compared to decreases of 2.2% during an El Niño event.  Similar results are found for the
other NH modes.

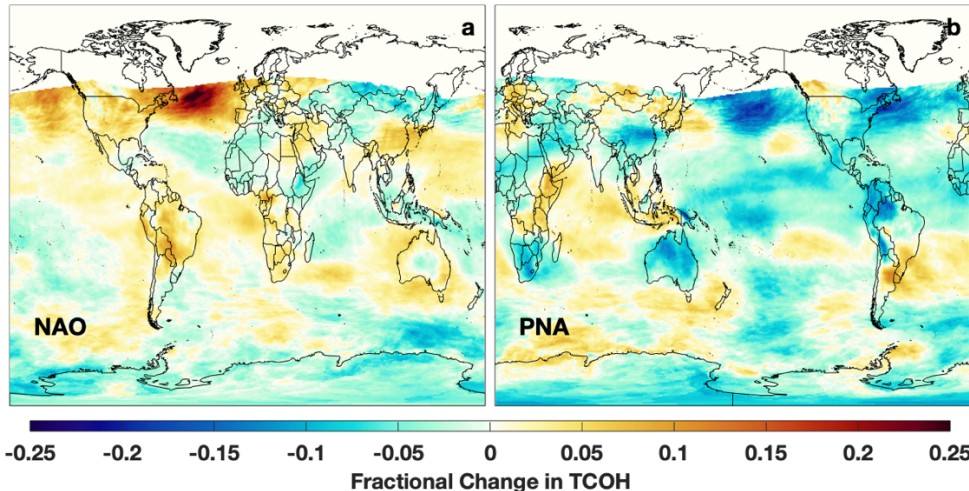

*Figure 20: Fractional change in TCOH for positive phases of the NAO (a) and PNA (b), defined as having an index greater than 0.4,*
*as compared to neutral events (index between -0.4 and 0.4).  Note that, for emphasis, the x-axis is shifted in panel b to center the*
*map over the Pacific Ocean.*
The effects of the monsoons on OH interannual variability are much more localized than for ENSO
and vary markedly among the different monsoons (Fig. S12).  For example, Figure S13a shows the
partial correlation coefficient (e.g., Sekiya and Sudo, 2012) of TCOH with the Australian monsoon,
taking into account the correlation of the Australian monsoon index with the MEI, which has an $r^2$
of 0.65 for DJF.  Correlation is almost exclusively restricted to areas near the Australian continent.
In this region, however, monsoons with an index in the 75th percentile or higher result in TCOH that
is 15-20% (up to 7 x $10^{11}$ molecules/cm$^2$) higher than for monsoons with an index between the 25th
and 75th percentile (Fig. S14).  These increases in OH column for the strongest monsoons are larger
in magnitude than typical changes associated with ENSO, although they are limited to a smaller
region, suggesting that the Australian monsoon can significantly perturb the local atmospheric
oxidative capacity.
In contrast, despite its larger scale, the Asian monsoon only shows correlation with TCOH over a
small portion of the subcontinent (Fig. S12b&d).  Correlations outside of the sub-continent region
result from the correlation between the Asian monsoon and ENSO.  Interestingly, this correlation is
only present during MAM and SON, not during JJA when the Asian monsoon is at full strength.
Lelieveld et al. (2018) have shown using *in situ* observations that upper tropospheric OH is
increased during the Asian monsoon.  The lack of correlation demonstrated here suggests that the
model is not accurately capturing the chemical variability within the monsoon anticyclone.  The
correlation with the monsoon index for MAM and SON could result from interannual variability in
the start and end of the monsoon.  Since these seasons are at the fringe of the monsoon, yearly
variations in the start and end date would lead to larger variability than that seen during JJA, when
the monsoon is active every year.
The IOD also shows a strong relationship with OH, although due to its annual cycle, the relationship
is only present during SON (Fig. 5d).  Taking into account the correlation between ENSO and the
IOD ($r^2$ = 0.30), the partial correlation between the Dipole Mode Index (DMI) and TCOH becomes
mostly restricted to the western Indian Ocean (Fig. S13b), where TCOH is anticorrelated with the
DMI, resulting in decreases in TCOH on the order of 10% (about 1.5 x $10^{11}$ molecules/cm$^2$).  During
the positive phase of the IOD, the Indian Ocean basin exhibits a Walker-type circulation with
anomalous surface easterly winds and increased convection in the region that exhibits
anticorrelation between OH and the DMI.  This region is also characterized by an anticorrelation
between the DMI and OH production from the NO + HO$_2$ reaction despite a positive correlation with
lightning NO emissions, analogous to the relationship between NO and ENSO in the equatorial
Pacific.  This suggests that the anticorrelation between TCOH and the DMI in the eastern Indian
Ocean is being driven by changes in NO transport from this Walker-type circulation.  More work is
needed, however, to prove this relationship, as the correlations between OH production from the
NO + HO$_2$ reaction and the DMI do not meet our stated statistical significance criteria.
**6.2 Simulated OH and the NH Climate Modes, Monsoons, and the IOD in the CCMI models**
The MERRA2 GMI and the CCMI simulations exhibit nearly identical spatial relationships between
TCOH and the NH climate modes and the IOD, demonstrating that these relationships are robust
among multiple models.  For example, all 5 models show two broad regions of correlation between
the NAO and TCOH, corresponding to the dipole pattern of the NAO (Fig. 21a).  Similar agreement is
found for the other NH modes (Fig. S15).  Likewise, most models show the same pattern of
correlation between the IOD and TCOH (Fig. 21b), consistent with their agreement for ENSO since
the two modes are closely related.

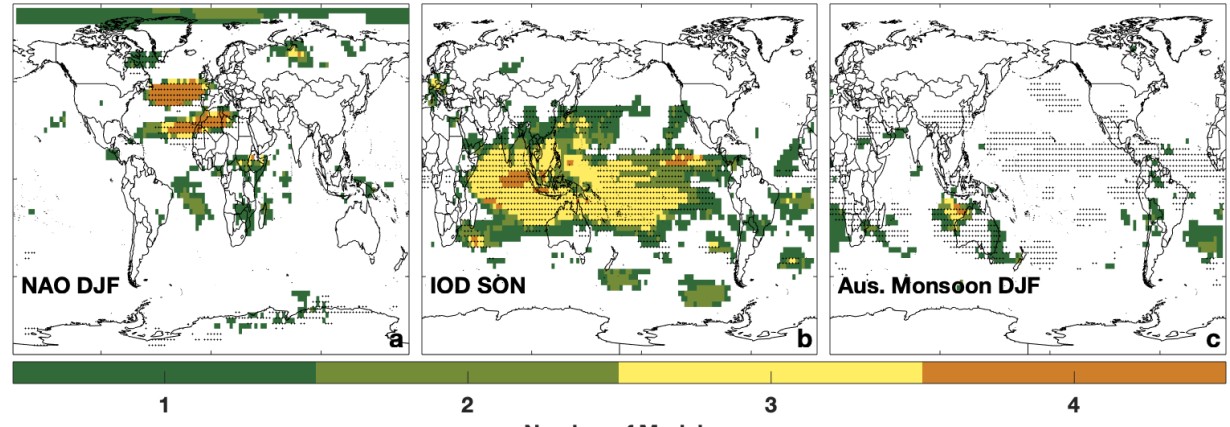

*Figure 21: Same as Figure 19 except for the NAO (a) the IOD (b), and the Australian Monsoon (c).  The NAO and Australian*
*monsoon are shown for DJF and the IOD for SON..*
In contrast to the other modes of variability, the relationship between TCOH and the different
monsoons varies widely among the models.  Agreement is highest for the Australian monsoon (Fig.
21c), where most models see correlation off the northwestern coast of the continent.  For the other
monsoons considered here, there is no consistent relationship with OH, with MERRA2 GMI being
the only model showing correlations with most monsoons (Fig. S16).  While models and
observations have shown the monsoons can change OH abundance, particularly in the UFT
(Lelieveld et al., 2018), the lack of correlation among the models suggests either that those changes
are not highly variable from year to year or that not all models capture the mechanisms behind
monsoon influence on OH, such as convective lofting of OH precursors.
**7.0 Conclusions**
Because of limited *in situ* observations and inter-model differences, there is significant uncertainty
in the processes driving interannual OH variability, despite its importance in controlling the
removal of many atmospheric trace gases.  Here, we have explored the relationship between OH
and multiple modes of climate variability, including ENSO, the IOD, NH modes of variability, and
monsoons in order to understand how these large-scale dynamical features influence OH through
control of its dynamical and photochemical drivers.

Using output from the MERRA2 GMI simulation, we have shown that during DJF, when considered
together, these climate features can explain a portion of OH variability over approximately 40% of
the troposphere by mass.  ENSO is the dominant mode of variability in all seasons except for JJA and
can explain 20 – 30 % of the spatial variance in TCOH and results in an average decrease in global,
mass weighted OH of 2.2% during El Niño events.  Effects from the other modes of variability
considered here are more limited in spatial scale but can strongly alter the atmospheric oxidative
capacity on the local scale.  For example, changes in TCOH for the NAO, IOD, and Australian
monsoon can reach 0.5, 1.5, and 7 x $10^{11}$ molecules/cm$^2$, respectively, compared to 2 x $10^{11}$
molecules/cm$^2$ for ENSO.
Changes in OH with ENSO are driven by different processes in the upper and lower troposphere.  In
the PBL, where OH production is dominated by the reaction of $O^1D$ with water, changes in the
distribution of these species leads to a positive correlation between OH and ENSO.  Increases in
$H_2O_{(v)}$ during El Niño are associated with increased convection and warmer SSTs, while increases in
$O^1D$ result from increased horizontal advection of $O_3$ in the western Pacific and increased
photolysis rates resulting from reduced stratospheric $O_3$ in the eastern Pacific.  In the upper
troposphere, NO controls the OH abundance over the tropical Pacific.  In much of the region,
decreases in lightning NO production correspond to decreases in total NO, and thus OH.  In the
equatorial region, however, increases in lightning NO production are offset by other processes,
potentially including transport due to changes in the Walker Circulation.  Further work is needed to
determine the relative importance of these two factors in controlling OH in the region during El
Niño and La Niña events.
Absolute changes in OH concentration during El Niño and La Niña events in the LFT and, to a lesser
extent, the MFT were limited by competing effects from changes in the $O^1D$ + $H_2O$ and NO + $HO_2$
reactions.  As a result, ENSO only explains 11.7% of the variability in the LFT and is associated with
the second EOF.  Because OH variability in the LFT drives variability in the $CH_4$ lifetime, which
showed limited response to ENSO variability, further research is warranted to understand the
dominant mode of OH variability at this level, including any impacts in emissions trends, which
appear to be the dominant mode of variability in the PBL.
The relationship between the individual climate modes seen in MERRA2 GMI is also seen in the
majority of the CCMI models, suggesting that the relationship between the modes and OH is robust.
4 of the 5 models examined here show similar relationships between ENSO and TCOH for all
seasons except MAM, and three of those models suggest that ENSO is the dominant mode of OH
variability in DJF, responsible for between 30 and 50% of total spatial variance.  Similar agreement
is found for the NH modes of variability and the IOD, while there is little agreement among models
between the relationship of the individual monsoons and OH.
Despite the agreement among models in the importance of the driving factors of OH variability,
there is still a lack of observations to demonstrate that the models are accurate.  We have shown
here that satellite observations of $H_2O$, CO, lightning flashes, and, to a lesser extent, $NO_2$ are able to
capture the respective variability of each variable as well as the relationship with ENSO, in excellent
agreement with the model simulation.  While further understanding of the relationship between
these species and ENSO is needed, the results presented here suggest that combining the
observations of OH drivers and the various climate modes could lead to additional methods to
constrain OH from space.
**Data Availability**

All output from MERRA2 GMI is publicly available at https://acd-ext.gsfc.nasa.gov/Projects/GEOSCCM/MERRA2GMI/.  Output from the EMAC, MRI, and CHASER models is available from the Centre for Environmental Data Analysis at http://data.ceda.ac.uk/badc/wcrp-ccmi/data/CCMI-1/output.  Output from WACCM are available at http://www.earthsystemgrid.org.  Satellite data are available at https://disc.gsfc.nasa.gov.  Data from the ATom campaign are located at https://espoarchive.nasa.gov/archive/browse/atom.

**Author Contributions**

DCA performed the analysis and wrote the manuscript.  All authors contributed ideas for the analysis and helped revise and improve the manuscript.

**Competing Interests**

The authors declare no competing interests.

**Acknowledgements**

The authors acknowledge funding from the NASA ACMAP program (16-ACMAP16-0027).  We acknowledge the modeling groups for making their simulations available for this analysis, the joint WCRP SPARC/IGAC Chemistry-Climate Model Initiative (CCMI) for organizing and coordinating the model data analysis activity, and the British Atmospheric Data Centre (BADC) for collecting and archiving the CCMI model output.

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
