# Peer review of "Spatial and temporal variability of the hydroxyl radical: Understanding the role of large-scale climate features and their influence on OH through its dynamical and photochemical drivers"

_Atmospheric Chemistry and Physics, 2020_

## Referee Comment (RC1) · Anonymous Referee #1 · 12 Jan 2021

**Review of Anderson et al., 2020**

**Major Comments:**
The goal of this study is to determine the relationship between tropospheric OH and ENSO, Northern Hemispheric modes of variability, the Indian Ocean Dipole, and monsoons. The authors present an analysis of one main model (GEOSCCM) evaluated with aircraft and satellite observations, to make the case that their model can be used for this purpose, and show that their findings have some similarities with the results from four CCMI models. This work finds that multiple modes of climate variability, including ENSO, can explain OH variability over approximately 40 % of the globe. The authors also find that OH mainly changes in the boundary layer and upper troposphere, and the mid troposphere is less impacted by different climate modes. I have several key concerns. First, the analysis of the model with aircraft and satellite observations is lacking in clarity, and possible technical approach, depending on the answers to my specific comments below. As an example, given the difficulty the model has in representing ATom OH observations in outflow from South America and New Zealand, changes over these regions should be more carefully discussed in Section 5. The authors might also discuss why they do not use any aircraft observations over land. In addition, a key finding of this paper is the finding that lightning $NO_x$ is not a main driver of the OH-ENSO relationship, in contrast to Turner et al., 2018. The authors need to improve their support of this argument. Finally, I think the authors are missing an opportunity to address the causes of the wide variety in OH and methane lifetime across models that they point out in their introduction. While the authors say that there is agreement across models in the importance of the driving factors of OH variability, for example only three models show that ENSO is important in DJF. This manuscript is appropriate for ACP and will represent a strong contribution to the field after the major revisions to the analysis described below.

**Specific Comments:**
Page 2, line 58 – It is incorrect to say that formaldehyde only comes from CH4 oxidation, please clarify what you mean here.

Page 3, line 135 – Could you clarify what you mean by detrend? Are you removing the seasonal cycle?

Page 3, line 143 – Could you describe whether the climate features you take from NOAA (MEI, DMI, the Northern Hemispheric modes) are well represented in the models so that you don't have to recalculate them from the model output?

Page 4, line 168 – Again please describe the method here a little better. What linear fit are you subtracting? What trend are you removing? Why do you have to divide by the standard deviation?

Page 5, line 202 – It might be clearer to say that "over the tropical Pacific area, …. emissions vary by up to 50 % over the time period studied."

Page 6, line 268 – Are you including the land crossings, or only using ocean data? The OH points that degrade the model/observation correlation seem like they must have some defining characteristic – if not over land, than maybe some other type of plume?

Page 6, line 275 – Per my comment above, would it not be more appropriate to average the ATom observations to the model output resolution, rather than interpolate to ATom? Otherwise, this is not an apples to apples comparison. There is no way the model will compare well to 5-min data when it is output at hourly resolution

Page 6, line 279 – I would prefer Fig S2 to be part of Figure 1. It seems arbitrary which is shown in the main text, and they are both important, particularly since Fig. S2 doesn't shown the same OH outliers in the SH as Figure 1. It would also be useful if they were on the same scale.

Page 7, line 285 – I see that the outliers are somewhat explained here as being driven by continental outflow from South America and New Zealand. Does this imply errors in model $NO_x$ in these regions? Or missing model OH recycling from biogenic VOC oxidation? Can ATom shed some light on this? Just stating that the correlation improves when those points are removed does not improve confidence in the ability of the model to simulate OH.

Page 7, line 293 – To my knowledge, it has not been shown that fixing the CO bias improves model OH biases. It is true that the low CO bias in the NH is well known, and the model OH bias is well known, but not that the OH bias is due to low CO. Please clarify your point here.

Fig. 2 caption, please explain what you mean by "satellite lifetime?"

Page 7, line 324 – Again it would be easier to read this if Fig. S3 was part of Fig. 2. It seems important enough to the discussion to warrant being in the main text.

Page 7, line 324 – You say that "overestimates of column CO, averaging 45%, in the SH corresponding with areas of biomass burning." Is that really true in SH winter (Fig. S3)?

Page 8, line 341 – Are you re-calculating the OMI $NO_2$ columns using the model a priori to give you an apples to apples comparison, per Lamsal et al., 2014 (doi:10.5194/acp-14-11587-2014)?

Page 10, line 408 – You say, "The relative importance of the individual reactions is similar during neutral and La Niña years (not shown) and is in agreement with previous model studies (e.g. Spivakovsky et al., 2000)." I am confused since you show in Fig. 4 that ENSO is correlated with TCOH, mainly in the tropics. How can this be if the relative importance of R1-4 is the same during neutral and La Nina years?

Page 10, line 413 – You do not use the term 'UFT' in the supplement (Fig. S4). Please be consistent in your terminology.

Page 12, line 456 – Does this mean that the variability shown in Fig 6a is driven mainly by changes in DJF as shown in Fig. 7a? If so, should Fig. 6 be shown seasonally similar to Fig. 7?

Page 14, line 514 – I don't understand the plot of satellite $NO_2$ against the MEI for Fig. 8. You say that "OMI data are insufficient," but show a plot anyway. If you aren't going to talk about it, maybe don't bother showing it. If you are filtering out noise from the $NO_2$ retrieval, and this leaves you only with $NO_2$ over land and outflow regions, why do you show ocean values for $NO_2$ in Fig. 2?

Figure S6 – Should the caption say, "model" $NO_2$ to differentiate from satellite?

Page 14, line 535 – Is there a plot somewhere for the sentence "Changes in the PBL during La Nina…?"

Pate 14, line 547 – You might want to consider the mass-weighted concentration of OH more consistently throughout the paper, and possibly the methane lifetime as well. This might provide an additional perspective on how different climate indices change the atmospheric oxidation capacity and address the variability in methane lifetime across models that you describe in the introduction.

Page 15, line 545 – According to Table 1, the $r^2$ with ENSO in JJA is less than 0.25, this should be pointed out.

Page 15, line 551 – Europe looks like it has decreases not increases, but it is hard to tell with the way the plot is centered. Please double check.

Page 17, line 558 – Do you mean "These increases in **j**O1D", not "These increases in O1D?" Also, does the model show that decreased stratospheric ozone is the driver behind the increase in photolysis?

Page 17, line 621 – It is surprising to me that if stratospheric ozone impacted jO1D in the PBL, it wouldn't also be important in the UFT. Please clarify.

Page 18, line 638 – I don't think the reference to Fig. S7 is correct, did you mean to refer to a different figure?

Page 18, line 645 – H2O(v) is Fig. 8a, not Fig. 8c.

Page 18, line 646 – It seems that there is a figure, Fig. 16, that you should refer to here in your discussion of lightning.

Page 19, line 652 – I don't understand your discussion of Turner et al. (2018). If I understand Turner et al. Fig. 4, correctly, there is more lightning during La Nina, but less during El Nino and thus La Nina is positively correlated with increases in OH. If I understand your Fig. 15 correctly, if NO goes down as MEI goes up, then NO goes up as the MEI goes down, meaning more lightning during a La Nina. This actually agrees with the findings of Turner et al. Please clarify if I am missing something here. If I am wrong, you will definitely need to better clarify your statement about biomass burning and be more specific about how interannual variability in emissions could have changed your results vs. Turner et al., 2018. For example, maybe you need a correlation plot of biomass burning emissions with MEI.

→ Page 19, line 667 – It seems that part of the argument is given here. Again, I would suggest that a biomass burning correlation plot or something similar would strengthen your argument. It does seem unlikely though that biomass burning would have such a large impact on the UFT, this definitely needs more discussion.

Page 21, line 740 – What is going on such that EMAC has no correlation in the column, but a very strong correlation in the UFT?

Figure 18 – While you focus on the agreement in your discussion of this figure, the discrepancies are actually quite large considering that all 5 models do not agree on the relationship of ENSO to TCOH in most cases. While you say it is beyond the scope of the paper on Page 22, line 748, given your analysis on the drivers of the ENSO to TCOH relationship from MERRA2 GMI, can you suggest areas of focus that might help us understand the huge model to model variability on OH and methane lifetime that you discuss in the introduction (e.g., Voulgarakis et al., 2013;Nicely et al.,;Zhao et al., 2019)?

Page 22, line 767 – It seems like Figure S13 is more appropriate for the main text.

Page 22, line 768 – Here you do talk about mass-weighted OH. I will just reiterate that it could be extremely useful to calculate changes to mass-weighted OH, or even better, changes to the methane lifetime in El Nino vs. La Nina.

Page 22, line 789 – Can you check the model output during JJA to see whether there is really no interannual variability? It does seem more likely that the model is not accurately capturing the chemical variability within the monsoon anticyclone.

Page 23, line 819 – Could this not also be that most models fail to capture the mechanism behind the monsoon impact on OH? If it is due to convectively lofting pollution above the monsoon clouds, then it seems very likely that all models have difficulty with this. Could you look to see whether your model has expected elevated levels of pollution during the monsoon in the UFT that might drive increased OH as described by Lelieveld et al., 2018?

Page 24, line 832 – Could you say something as well about the average decrease in the methane lifetime (using the methane concentrations from the model)?

Page 24, line 844 – This statement about lightning emissions is very important, and definitely needs more support in the prior text. I don't think that "increased convective lofting of low NO air from near the surface and advection of air with lower reactive nitrogen than during neutral years" has been well supported.

**Technical Corrections**

Page 7, line 288 – "a NMB", not "an NMB"

Page 15, line 546 – I think it should be "ENSO-related".

---

## Referee Comment (RC2) · Anonymous Referee #2 · 13 Feb 2021

The manuscript acp-2020-1192 evaluates the drivers of interannual variability in hydroxyl radical (OH) throughout the troposphere. The authors primarily use model output from a 38-year chemical transport model to calculate OH variability and correlate this spatially and temporally with various climate modes, with specific focus on ENSO. They also compare results to other atmospheric chemistry models and where possible

satellite data to test the robustness of their conclusions. This is a detailed, careful, and well presented analysis that adds to our understanding of OH science and should be published in ACP once the comments below have been addressed.

GENERAL COMMENTS

Methane feedbacks: There are known feedbacks between ENSO, methane, and OH. However, according to Section 2.2 (lines 203-204), methane in the model is a fixed boundary condition, which I understand to mean that it wouldn't vary in response to ENSO or other conditions. Therefore some component of the ENSO-driven OH variability may be missing in the simulation (this may be true for other modes too, e.g. Australian monsoon). It would be worth a brief discussion somewhere of this and the implications for the conclusions.

Satellite comparisons (Section 3.2): As far as I can tell, the model comparisons to satellite data have NOT taken into account the retrieval process used in the satellite data (e.g., application of averaging kernels and a priori for CO / shape factors and scattering weights for NO). This introduces an element of bias into the comparisons – even if the model was 100% accurate, the values retrieved by the satellite would not match the model (because the satellite retrievals are not a perfect observing system). While this is an understandable choice for the scale of analysis in this paper, this needs to be clearly discussed. It also means that the comparisons should be considered more qualitative than quantitative (e.g. show of the quoted biases are probably not accurate), and some of the conclusions from this section should be stated less strongly.

SPECIFIC COMMENTS BY LINE NUMBER

79-91: The relationships between a variety of climate modes and CO anomalies quantified by Buchholz et al. (2018; https://doi.org/10.1029/2018JD028438) would be relevant to include in this discussion. Also, the last sentence notes observed relationships between NAO and CO long-range transport, but other work has shown CO long-range transport also changes with other climate modes (e.g. with ENSO, see Fisher et al.,

2010; https://doi.org/10.5194/acp-10-977-2010).

143-162: I suggest including a table that lists all the modes and monsoons included in the analysis and a column that indicates their source (for the modes) or the fact that they are calculated (for the monsoons). Right now I think it's a bit hidden that the monsoons are model-dependent while the modes are not (and I wonder if this has any influence on the fact that you tend not to find agreement between the models in the monsoon analysis in Section 6).

203-204: More details of the methane concentration implementation would be useful here. What is the source for the methane values? At what latitudinal and time resolution are they input to the model? Are the specified concentrations applied throughout the troposphere, or only at the surface then allowed to advect freely?

229-231: What biomass burning emissions are used in the CCMI models, and are they consistent with the GEOSCCM biomass burning emissions?

325: I'm somewhat surprised to see the overestimates in the SH in JJA linked to biomass burning, as this is not peak burning season in the SH, and the spatial patterns within the SH continents don't look like the areas of primary dry season burning. Is there other evidence to support the conclusion that this is a biomass burning signal?

404: How are El Niño "events" defined? And what is the timescale used? I think in some places the terminology is "events" and in other places it is "years" – are there sub-annual events, or is each year classified in its entirety? This also needs addressing in the Figure 6 caption, which describes a difference between "El Niño events and neutral years" – if events are sub-annual, how does this work?

478: Does the Percent Variance in the table (and quoted throughout the text) refer to the spatial variance? It would be useful if this were clarified somewhere in the text, as it's a little confusing to see such large disconnect between the percent variance explained and the r^2.

513-514: "Analysis of vertical winds..." – it's not clear to me how Fig. S6 supports this conclusion (and I find it hard to tell what is going on in Fig. S6). To clarify would probably require more discussion – but as the paper is already very long with a huge number of figures between the main text and the supplement, I would suggest just cutting this one.

544: "the 2nd EOF for the PBL" – it would be useful to have an equivalent figure to Figure 7 included in the supplement to show the 2nd EOF for the PBL, since it is the most correlated EOF here.

583-584: This sentence needs a reference

654-656: I am confused and think more discussion is warranted here. The statement is that the difference in the role of lightning NO emissions could be due to "differences in the handling of biomass burning emissions in the two models", but only the MERRA2 GMI biomass burning emissions are explained. How were the biomass burning emissions handled in the Turner et al. study? Second, why would the lightning NO emissions change in response to the biomass burning emissions? Or is the argument that the Turner et al. study was missing variability in biomass burning and so incorrectly attributed the variability to lightning NO? Please explain.

715: In Fig 18 and other similar figures, I'm not sure of the validity of including the MERRA2 GMI model that is the focus of the paper in the model count. What would be more useful would be to see how many of the independent CCMI models reproduce the relationship seen in MERRA2 GMI. I think this could be done by changing these figures to have only the 4 models in the color scale, but then using e.g. hatching to overlay where MERRA2 GMI had a significant correlation. This would be a clearer comparison of the robustness of the MERRA2 GMI relationships already identified.

799-800: "TCOH is anticorrelated with the DMI" – any ideas why?

TECHNICAL COMMENTS BY LINE NUMBER

29: "Reductions in OH due to ENSO" – please change ENSO to El Niño as ENSO would imply OH is reduced during both phases.

195: Please specify the version of MEGAN used.

263: I would suggest turning this into a 4-panel figure that combines the existing Figure 1 with the existing Figure S2 – the ATom 1 comparison is also useful to see without having to flip to the supplement.

286: "When data from these regions are omitted" – it would be useful to identify the "omitted" data points in the plots shown in Fig. 1 (e.g. with a different shape, filled circle, etc.).

292-293: It would be worth referencing the more recent work of Travis et al. (2020; https://doi.org/10.5194/acp-20-7753-2020) here.

319: "boreal winter" – specify months in parentheses.

376: "In JJA, the climate modes" – please clarify what this refers to. . . Is it just the NH modes? All climate modes (NHmodes + ENSO + IOD)? Or all drivers (NHmodes + ENSO + IOD + monsoons)?

407-408: "We focus our analysis on DJF. . . ." – the text would be clearer if this sentence came earlier, before the current sentence starting on line 405 "Figure 5 shows. . ."

425: Duplicated 5.1.1, this should just be 5.1

483-484: "with greater than" – please change to "with NO2 greater than"

518-519: "The dominant OH sink. . ." – this sentence would fit better with the earlier discussion of Figure 5 and the dominant OH source pathways.

527: "Same as Figure 6" –> "Same as Figure 6ab"

536: "smaller than in the UFT (Fig. 10)" – it is odd to mention the UFT in this context before it has been discussed. I suggest removing this part and just keeping the

comparison to TCOH here.

557: "Figure 11 shows…" – the wording is unclear both here and in the caption. I think in both cases, the correlation is with the MEI, but in one case it is MEI vs. OH production from H2O+O1D and in the other it is MEI vs. total OH production rate. If this is correct, I would suggest clarifying the sentence to start "Figure 11 shows the correlation of MEI against both OH production from this reaction and the total OH production rate for the PBL."

577: Here and elsewhere, JO1D → J(O1D)

638: "Fig. S7" – should be "Fig S6." (although I have suggested earlier that this figure be cut)

643-644: "lightning emissions … does not" → "lightning emissions do not"

644-651: There should be a reference to Fig. 16 somewhere in the text of this section.

708-709: "the CCMI models" – it would be useful to remind the reader here how many CCMI models are being included (4 I think?); e.g. "the 4 CCMI models…"

765-767: I find this long sentence difficult to interpret. I would suggest splitting into two sentences, one about the NAO and one about the PNA.

774: "vary markedly among the different monsoons (Fig. 4)" – I find this very hard to see from Fig. 4. I would suggest a separate figure in the supplement that shows just the monsoon correlations to aid this discussion. Same goes for the reference to Fig. 4b&d in line 786.

811: Suggest swapping the order of panels b and c to reflect the order in which they are discussed in the text.

Throughout: If I understand correctly, the main model is sometimes referred to as GEOSCCM and sometimes referred to as MERRA2 GMI - please make the name of the model consistent throughout.

---

## Author Comment (AC1) · 19 Mar 2021

We thank the reviewer for their comments, which have helped to improve the manuscript. Responses to comments are shown below in red. Page and Line numbers refer to the track changes version of the resubmitted manuscript.

The manuscript acp-2020-1192 evaluates the drivers of interannual variability in hydroxyl radical (OH) throughout the troposphere. The authors primarily use model output from a 38-year chemical transport model to calculate OH variability and correlate this spatially and temporally with various climate modes, with specific focus on ENSO. They also compare results to other atmospheric chemistry models and where possible data to test the robustness of their conclusions. This is a detailed, careful, and well-presented analysis that adds to our understanding of OH science and should be published in ACP once the comments below have been addressed.

**GENERAL COMMENTS**

Methane feedbacks: There are known feedbacks between ENSO, methane, and OH. However, according to Section 2.2 (lines 203-204), methane in the model is a fixed boundary condition, which I understand to mean that it wouldn't vary in response to ENSO or other conditions. Therefore some component of the ENSO-driven OH variability may be missing in the simulation (this may be true for other modes too, e.g. Australian monsoon). It would be worth a brief discussion somewhere of this and the implications for the conclusions.

We now include a paragraph at the end of Section 2.2 to discuss how CH4 is treated in the MERRA2 GMI simulation, and the implications for our conclusions (Page 6, Line 269-280):

Because CH4 is specified as a boundary condition, the model does not capture feedbacks (e.g., wetland or wildfire emissions) between CH4 emissions and climate modes beyond the extent to which these manifest in the observed methane surface concentrations. ENSO, for example, is known to affect atmospheric CH4 concentrations through changes in emissions from wetlands (Zhang et al., 2018;Melton et al., 2013) and biomass burning (Worden et al., 2013), although there is uncertainty in the magnitudes of these effects (Melton et al., 2013). On the global scale, however, these ENSO-induced changes in emissions do not significantly perturb background CH4. For example, during the 1997/98 ENSO event, one of the largest on record, CH4 grew at a rate of approximately 15 ppbv/yr on top of a background on the order of 1700 ppbv (Nisbet et al., 2016). Because of this small perturbation and the dominance of CO as the primary OH sink over much of the globe (see Section 5.0), it is unlikely that the relationship between climate modes and OH would differ significantly with the inclusion of direct methane emissions in the simulation.

Satellite comparisons (Section 3.2): As far as I can tell, the model comparisons to satellite data have NOT taken into account the retrieval process used in the satellite data (e.g., application of averaging kernels and a priori for CO / shape factors and scattering weights for NO). This introduces an element of bias into the comparisons –even if the model was 100% accurate, the values retrieved by the satellite would not match the model (because the satellite retrievals are not a perfect observing system). While this is an understandable choice for the scale of analysis in this paper, this needs to be clearly discussed. It also means that the comparisons should be considered more qualitative than quantitative (e.g. show of the quoted biases are probably not accurate), and some of the conclusions from this section should be stated less strongly.

We agree that we should have been clearer in the methodology used to compare the model and satellite. For the MOPITT CO comparison, we do convolve the model output with the averaging kernel/a priori from the CO retrieval, so quantitative comparisons between the satellite and model are possible. For OMI NO2, we have not applied any scattering weights or shape factors. We do not anticipate this leading to major discrepancies, however. The shape factors used for the OMI NO2 retrieval are derived from a GEOSCCM model run with a similar setup to that described here, including using the GMI chemical mechanism and MERRA2 meteorology. So, differences in the shape factors used in the satellite retrieval and those determined from the MERRA2 GMI run described in this paper will be a minimum. We include the following discussion (Page 9, Lines 421-431):

For comparison of the satellite retrievals to MERRA2 GMI, we use monthly fields of the model variables output at the satellite overpass time. For CO, where averaging kernel and *a priori* information are available for the Level 3 MOPITT data, we convolve the model output with these variables so that direct comparison between satellite and model are possible. While shape factors and scattering weights for the OMI NO2 retrieval are unavailable for the Level 3 data, shape factors for the OMI NO2 retrieval are determined from a similar setup of the GEOSCCM model, also

employing the GMI chemical mechanism and MERRA2 meteorology. Applying the satellite shape factors to the simulation discussed here would therefore not result in significant changes in the modeled NO2. Finally, for AIRS  $H_2O$ , averaging kernel information was unavailable for the Level 3 data, so numerical comparisons between satellite and model should be regarded as more qualitative than quantitative.

**SPECIFIC COMMENTS BY LINE NUMBER**

79-91: The relationships between a variety of climate modes and CO anomalies quantified by Buchholz et al. (2018; https://doi.org/10.1029/2018JD028438) would be relevant to include in this discussion. Also, the last sentence notes observed relationships between NAO and CO long-range transport, but other work has shown CO long-range transport also changes with other climate modes (e.g. with ENSO, see Fisher et al.).

We have added a reference to Buchholz et al. as follows (Page 2, Line 89-100):

In addition to this biomass burning relationship with ENSO, Buchholz et al. (2018) also noted relationships between tropical fire regions and the IOD as well as with the Tropical South Atlantic and Southern Annular modes.

Likewise, we have modified the final sentence of the paragraph to read (Page 2, Line 102-104):

Finally, climate modes can alter the long range transport of CO to the Arctic, through increased outflow from Europe (Li et al., 2002;Creilson et al., 2003;e.g. Duncan, 2004) and Asia (Fisher et al., 2010) for the NAO and ENSO, respectively.

143-162: I suggest including a table that lists all the modes and monsoons included in the analysis and a column that indicates their source (for the modes) or the fact that they are calculated (for the monsoons). Right now I think it's a bit hidden that the monsoons are model-dependent while the modes are not (and I wonder if this has any influence on the fact that you tend not to find agreement between the models in the monsoon analysis in Section 6).

We have included the suggested table in the paper and reproduce it here.

**Table 1:** Summary of the climate modes and monsoons considered in this work. The index used to characterize the mode, as well as the source of the index, is also indicated.

| Mode Type                 | Index                                                                                                         | Mode Type                     | Index                                                                       |
|---------------------------|---------------------------------------------------------------------------------------------------------------|-------------------------------|-----------------------------------------------------------------------------|
| El Niño Southern          | Multivariate ENSO                                                                                             | North Atlantic                | EOF of geopotential
height at 500 mbar
from NCEP
reanalysis (NOAA) |
| Uscillation               | Dinala Mada Inden                                                                                             | Oscillation                   |                                                                             |
| Dipole                    | (NOAA)                                                                                                        | East Atlantic                 |                                                                             |
| Asian Monsoon             | Model-specific
index calculated
from the difference
of zonal winds in
monsoon specific
regions | Pacific North
American     |                                                                             |
| South American            |                                                                                                               | East Atlantic                 |                                                                             |
| Monsoon                   |                                                                                                               | Western Russian               |                                                                             |
| North American
Monsoon |                                                                                                               | Scandinavian                  |                                                                             |
| South African
Monsoon  |                                                                                                               | West Pacific                  |                                                                             |
| North African
Monsoon  |                                                                                                               | East Pacific North
Pacific |                                                                             |
| Australian Monsoon        |                                                                                                               | Tropical Northern             |                                                                             |
| Western North             |                                                                                                               | Hemisphere                    |                                                                             |
| Pacific Monsoon           |                                                                                                               | mennisphere                   |                                                                             |

203-204: More details of the methane concentration implementation would be useful here. What is the source for the methane values? At what latitudinal and time resolution are they input to the model? Are the specified concentrations applied throughout the troposphere, or only at the surface then allowed to advect freely?

We have updated the text to more fully explain the methane constraints in the model. The text now reads (Page 5-6, Lines 250-267):

Methane concentrations are specified at the surface for 4 different latitude bands  $(90^{\circ}\text{S} - 30^{\circ}\text{S}, 30^{\circ}\text{S} - 0^{\circ}, 0^{\circ} - 30^{\circ}\text{N}, 30^{\circ}\text{N} - 90^{\circ}\text{N})$  at monthly resolution and advected throughout the troposphere. Methane data are from the NOAA Global Monitoring Division (GMD) surface network (Dlugokencky et al., 1994) and monthly values are interpolated from annual means.

229-231: What biomass burning emissions are used in the CCMI models, and are they consistent with the GEOSCCM biomass burning emissions?

We now include a brief discussion of the biomass burning emissions from CCMI and how they compare to the GFED4s inventory used in MERRA 2 GMI (Page 6, Lines 307 - 313):

Biomass burning emissions are from Granier et al. (2011), which incorporate a modified version of the RETRO inventory from 1980 – 1996 and GFEDv2 from 1997 – 2010 and are based on Lamarque et al. (2010). Monthly-averaged CO emissions from this inventory in Indonesia, where biomass burning emissions are strongly affected by ENSO (e.g., Duncan, 2003a), are highly correlated ( $r^2 = 0.79$ ) in time with the GFED 4s inventory used in the M2GMI simulation. Likewise, monthly-averaged CO emissions over Indonesia from the two inventories agree within 35%, on average.

325: I'm somewhat surprised to see the overestimates in the SH in JJA linked to biomass burning, as this is not peak burning season in the SH, and the spatial pat-terns within the SH continents don't look like the areas of primary dry season burning. Is there other evidence to support the conclusion that this is a biomass burning signal?

We agree with you, and the other reviewer, that biomass burning is likely not the cause of the JJA disagreement. A more likely explanation for the difference is due to errors in isoprene emissions and chemistry. We now say (Page 10, Line 4347-449):

These areas of high bias over South America likely result from the high bias in isoprene emissions, as discussed in Section 2.2, that would lead to unrealistically high *in situ* production of CO.

404: How are El Niño "events" defined? And what is the timescale used? I think in some places the terminology is "events" and in other places it is "years" – are there sub-annual events, or is each year classified in its entirety? This also needs addressing in the Figure 6 caption, which describes a difference between "El Niño events and neutral years" – if events are sub-annual, how does this work?

We agree that the term neutral year is confusing and now use "neutral event". In fact, El Niño, La Niña, and neutral events are all defined on the seasonal scale. We now formally define our terminology as follows (Page 4, Line 171-173):

We use monthly values of the ENSO multivariate index (MEI) (Wolter and Timlin, 2011) obtained from https://psl.noaa.gov/enso/mei and averaged to seasonal time scales. Here, ENSO-related events are defined according to the seasonally averaged MEI, where MEI > 0.5 is an El Niño event, MEI < -0.5 is a La Niña event, and an MEI value between 0.5 and -0.5 is a neutral event.

In addition we updated the caption to Figure 7 (formally Figure 6) to reflect the new terminology and changed language throughout the text from neutral years to neutral events.

478: Does the Percent Variance in the table (and quoted throughout the text) refer to the spatial variance? It would be useful if this were clarified somewhere in the text, as it's a little confusing to see such large disconnect between the percent variance explained and the r2.

Yes, we have updated all references to variance in the text to reflect that we are talking about the spatial variance.

513-514: "Analysis of vertical winds..." – it's not clear to me how Fig. S6 supports this conclusion (and I find it hard to tell what is going on in Fig. S6). To clarify would probably require more discussion – but as the paper is already very long with a huge number of figures between the main text and the supplement, I would suggest just cutting this one.

We have removed the figure and appreciate the desire to keep the paper a more manageable size.

544: "the 2nd EOF for the PBL" – it would be useful to have an equivalent figure to Figure 7 included in the supplement to show the 2nd EOF for the PBL, since it is the most correlated EOF here.

We agree that this is a good idea. We include the suggested figure, shown below, as Figure S6.

Figure 1: The 2nd EOF of PBL OH from MERRA2 GMI for DJF (a), MAM (b), JJA (c), and SON (d)

In addition, we have added the following text to the paragraph discussing EOFs in the PBL (Page 18, Line 770-771):

The spatial pattern of the 2nd EOF for PBL OH varies markedly across seasons (Fig. S6), with the largest signal over the tropical Pacific during DJF and MAM and over Indonesia in SON.

583-584: This sentence needs a reference

We have included a reference to Oman et al. (2011).

654-656: I am confused and think more discussion is warranted here. The statement is that the difference in the role of lightning NO emissions could be due to "differences in the handling of biomass burning emissions in the two models", but only the MERRA2GMI biomass burning emissions are explained. How were the biomass burning emissions handled in the Turner et al. study? Second, why would the lightning NO emissions change in response to the biomass burning emissions? Or is the argument that the Turner et al. study was missing variability in biomass burning and so incorrectly attributed the variability to lightning NO? Please explain.

We agree, with both you and the other reviewer, that we did not clearly articulate how the results of our work differ from that of Turner et al. We had intended to convey the idea that additional mechanisms, besides lightning NO production, likely lead to the modeled relationship between NOy species and ENSO in the equatorial Pacific, where the model shows a positive correlation with lightning NO and the MEI but an anticorrelation in NO (as well as other NOy species). Outside of the equatorial region, both lightning NO emissions and NO concentrations are anticorrelated with the MEI, suggesting decreased NO during El Niño, which is consistent with the decreased OH concentrations, and increased NO during La Niña. So our work is in agreement with Turner outside the equatorial region. We have updated this discussion as follows (Page 23, Lines 947-965):

This tri-pole correlation pattern between MEI and lightning, evident in both the satellite and model (Fig. 17) is in contrast to the relationship with NO (Fig. 16a) and other reactive nitrogen (NOy) species in the UFT. While the anti-correlation in NO is consistent with the changes in lightning NO emissions in some regions, in the equatorial Pacific band, NO decreases during El Niño events despite an increase in lightning NO emissions. This apparent discrepancy occurs because even though lightning NO increases by 100% or more over the equatorial Pacific during El Niño events in the model, the absolute difference is orders of magnitude lower than the accompanying changes over land. We conclude that the resulting NO perturbations over the equatorial Pacific latitudes are

dominated by mechanism other than the local lightning response, such as changes in the Walker Circulation and the associated transport of air originating over the continents. This mechanism is supported by the similar regression pattern of longer-lived species, such as HNO3 (Fig. **Error! Reference source not found.**c) and PAN (not shown), to NO in the UFT supports this idea, showing that transport of reactive nitrogen from other source regions, particularly lightning over South America, is likely reduced during El Niño events.

Our findings are broadly consistent with Turner et al. (2018), who found that increases in lightning NO emissions drive increases in OH during La Niña and, conversely, decreases in lightning NO emissions lead to OH decreases during El Niño. The results presented here suggest that in addition to this influence of lightning locally, other mechanisms, such as atmospheric transport of NOy species, also likely contribute to the relationship between ENSO and OH in the equatorial Pacific.

715: In Fig 18 and other similar figures, I'm not sure of the validity of including the MERRA2 GMI model that is the focus of the paper in the model count. What would be more useful would be to see how many of the independent CCMI models reproduce the relationship seen in MERRA2 GMI. I think this could be done by changing these figures to have only the 4 models in the color scale, but then using e.g. hatching to overlay where MERRA2 GMI had a significant correlation. This would be a clearer comparison of the robustness of the MERRA2 GMI relationships already identified.

Altering Figure 18 (now Fig. 19) to highlight agreement among the different CCMI models is a good idea. We have changed this figure, along with the Figures 19 (now Fig. 21), S16 (now S15), and S17 (now S16) to only include the CCMI models. We have included stippling to indicate the grid boxes where MERRA2 GMI also shows significant correlations between the different variables. We include a sample figure here.

---

## Author Comment (AC2) · 19 Mar 2021

We thank the reviewers for their comments and have addressed them in red below. Page and Line numbers refer to the tracked changes version of the documents.

Review of Anderson et al., 2020
Major Comments:
The goal of this study is to determine the relationship between tropospheric OH and ENSO, Northern Hemispheric modes of variability, the Indian Ocean Dipole, and monsoons. The authors present an analysis of one main model (GEOSCCM) evaluated with aircraft and satellite observations, to make the case that their model can be used for this purpose, and show that their findings have some similarities with the results from four CCMI models. This work finds that multiple modes of climate variability, including ENSO, can explain OH variability over approximately 40 % of the globe. The authors also find that OH mainly changes in the boundary layer and upper troposphere, and the mid-troposphere is less impacted by different climate modes. I have several key concerns. First, the analysis of the model with aircraft and satellite observations is lacking in clarity, and possible technical approach, depending on the answers to my specific comments below. As an example, given the difficulty the model has in representing ATom OH observations in outflow from South America and New Zealand, changes over these regions should be more carefully discussed in Section 5. The authors might also discuss why they do not use any aircraft observations over land. In addition, a key finding of this paper is the finding that lightning $NO_x$ is not a main driver of the OH-ENSO relationship, in contrast to Turner et al., 2018. The authors need to improve their support of this argument. Finally, I think the authors are missing an opportunity to address the causes of the wide variety in OH and methane lifetime across models that they point out in their introduction. While the authors say that there is agreement across models in the importance of the driving factors of OH variability, for example only three models show that ENSO is important in DJF. This manuscript is appropriate for ACP and will represent a strong contribution to the field after the major revisions to the analysis described below.

Specific Comments:
Page 2, line 58 – It is incorrect to say that formaldehyde only comes from CH4 oxidation, please clarify what you mean here.

We have changed this sentence to read (Page 2, Line 68-70):

> "Recent work has demonstrated that formaldehyde, a longer-lived species (hours) whose chemical production in the remote troposphere is often controlled by $CH_4$ oxidation, shows promise for inferring variability in OH columns over the remote atmosphere (Wolfe et al., 2019)."

Page 3, line 135 – Could you clarify what you mean by detrend? Are you removing the seasonal cycle?

For each grid box, we remove any linear trend in OH on a month-by-month basis to account for any changes in background values, as an increasing or decreasing background could affect the correlations. There is no need to remove the seasonal cycle as we perform separate correlations for each season. We have clarified the text (Page 3, Line 155-157) to read as follows:

> "To perform the regression, we first detrend the output on a monthly basis, removing any linear trend from each variable over the 1980 to 2018 period to account for changes in the background value."

Page 3, line 143 – Could you describe whether the climate features you take from NOAA (MEI, DMI, the Northern Hemispheric modes) are well represented in the models so that you don't have to recalculate them from the model output?

This is a good question. The Dipole Mode Index as calculated by NOAA is based on Sea Surface Temperatures (SSTs) from the Hadley SST dataset, which is the same dataset used in the CCMI model runs. While MERRA2 SSTs are assimilated from various sources, they agree well (within 0.2 degrees) with other reanalyses (Bosilovich, 2015). Because of the similarities in these datasets, the DMI should be nearly identical to analogously defined indices calculated from each of the models examined here. Likewise, the MEI is calculated using sea level pressure (SLP), SSTs, and surface level zonal and meridional winds from the JRA-55 dataset, which was also used to constrain the MRI model in this study. As demonstrated in Orbe et al (2020), there was good agreement in the

temperatures and zonal winds in the CCMI SD runs for the models constrained to the JRA-55, ERA-Interim, and MERRA reanalyses, while agreement in meridional winds was good, but not as robust. MERRA2 surface winds, both zonal and meridional, agree within less than 0.2 m/s in the tropics with both the MERRA and ERA-Interim datasets. Therefore, the MEI would agree with climate indices calculated directly from these models. Likewise, the MEI is highly correlated with other ENSO indices, such as the Nino3.4 index, calculated directly from SSTs, which as we mention above, are almost identical among the different models considered here. Finally, for the Northern Hemispheric modes, we do not have geopotential height at 500 mbar for all of the models considered here so would not be able to calculate the indices for each of the northern hemispheric modes, as defined by NOAA. For comparison, we have calculated an NAO index from MERRA2 GMI using the difference between SLPs at Iceland and the Azores. Correlation between this index and the NOAA derived NAO index agree with an $r^2$ of 0.79, suggesting that the NOAA index is able to describe the NAO in the model. Further evidence is given by the results shown in Figures 21a and S15, which show the relationship between OH and the various northern hemispheric modes for each model, where the areas of correlation correspond to the regions typically associated with each of the modes.

We have summarized this work in the paper (Page 4, lines 183-190):

> "The MERRA2 GMI (Section 2.2) and CCMI models (Section 2.3) included here are constrained or nudged to reanalyses data (MERRA, MERRA2, JRA-55, and the ERA-interim) which assimilate observed meteorology. The meteorological variables used to calculate the DMI and MEI, including sea surface temperature, sea level pressure, and zonal and meridional winds, agree well or are identical among the different reanalyses (Orbe et al., 2020;Bosilovich et al., 2015). Thus, climate modes in these models correspond to the NOAA indices. Likewise, indices for the NAO calculated from surface pressure from the models correlate well ($r^2$ of 0.79 or greater) with the NAO index calculated by NOAA."

Page 4, line 168 – Again please describe the method here a little better. What linear fit are you subtracting? What trend are you removing? Why do you have to divide by the standard deviation?

We have removed reference to dividing by the standard deviation as this does not affect our results, although this is necessary for some forms of extended EOF analysis not performed here. We have edited the description as follows (Page 5, line 214-216):

> "To perform the analysis, OH fields for each grid box were detrended by subtracting a linear fit to the time series over the 1980 to 2018 period to account for changes in background associated with long-term trends in OH."

Page 5, line 202 – It might be clearer to say that "over the tropical Pacific area, …. emissions vary by up to 50 % over the time period studied."

We have made this change.

Page 6, line 268 – Are you including the land crossings, or only using ocean data? The OH points that degrade the model/observation correlation seem like they must have some defining characteristic – if not over land, than maybe some other type of plume?

We include all OH and CO observations from ATom, which we now clarify in the paper. We describe the model outliers more thoroughly in the response to Page 7, Line 285 below.

Page 6, line 275 – Per my comment above, would it not be more appropriate to average the ATom observations to the model output resolution, rather than interpolate to ATom? Otherwise, this is not an apples to apples comparison. There is no way the model will compare well to 5-min data when it is output at hourly resolution

We spent some time debating the best way to do a measurement/model comparison, and each way presents its own set of complications. If the observations were made at a constant location, we would agree with your suggested approach. Because of the campaign sampling strategy, however, averaging the ATom observations to an hourly time scale averages over space rather than time and is still not an apples-to-apples comparison. For example, over an hour period, an ATom flight could complete approximately two vertical profiles spanning from ~250 hPa to ~950 hPa. Averaging data over this period could involve multiple air masses with distinct features, none of which would

be represented by an average. These averaged data would therefore not be comparable to the model output for the averaged latitude, longitude, and pressure of the sampling interval. Similar problems would be present if the plane were only travelling horizontally. While we could use observations averaged over a shorter time scale near the model output time (observations made within 2.5 min of the model output time, for example) this would greatly limit the number of data points for comparison. We also note that this method of comparison has been used for OH in the remote atmosphere in previously published work (e.g., Nicely et al, 2016).

Nicely, J. M., et al. (2016). "An observationally constrained evaluation of the oxidative capacity in the tropical western Pacific troposphere." Journal of Geophysical Research: Atmospheres **121**: 7461-7488.

Page 6, line 279 – I would prefer Fig S2 to be part of Figure 1. It seems arbitrary which is shown in the main text, and they are both important, particularly since Fig. S2 doesn't shown the same OH outliers in the SH as Figure 1. It would also be useful if they were on the same scale.

We had included ATom 2 as a figure in the main text because, for the most part, the work focuses on relationships during DJF, and limited the ATom 1 figure to the supplement to help control length of an already long paper. We have combined the two figures, which we show here, making sure that the axes have the same scale for both ATom deployments.

[Figure]

Page 7, line 285 – I see that the outliers are somewhat explained here as being driven by continental outflow from South America and New Zealand. Does this imply errors in model NO$_x$ in these regions? Or missing model OH recycling from biogenic VOC oxidation? Can ATom shed some light on this? Just stating that the correlation improves when those points are removed does not improve confidence in the ability of the model to simulate OH.

This is a good question. It is difficult to evaluate the cause of differences between observations and the model because hourly resolution output for MERRA2 GMI is only available for a small number of species, which do not include NO$_y$ constituents or any VOCs. We have looked at the correlations in Figure 1a colored by different observed species, such as NO which is shown below, but there is no obvious relationship between the data points underestimated by the model and any of these species. Particularly off the coast of South America, it is possible that this underestimate is related to incorrect isoprene chemistry and emissions in the model, as there is a known high bias in isoprene emissions in the model, resulting in unrealistically low OH over parts of South America. We note this in the text as follows (Page 8, Line 374-378):

> The limited model output at hourly resolution does not allow for a determination of the cause of this disagreement in continental outflow regions. In the case of South America, however, a known high

bias in modeled isoprene, resulting in extremely low OH over the Amazon, is consistent with the disagreement between the simulation and observations.

Because the measure/model disagreement is limited to these continental outflow regions and the majority of the correlations discussed in the paper are in the remote atmosphere, in regions unaffected by fresh continental emissions, we do not explore the causes of the disagreement in more detail. We also note, that outside of these outflow regions, the model agrees with the measurements within measurement uncertainty. We have updated the text to read (Page 9, Line 400-403):

> This lack of agreement does not significantly affect the results discussed in this work, as the majority of the relationships found between OH and modes of climate variability discussed in Sections 4 and 5 are centered in the remote atmosphere.

[Figure]

Page 7, line 293 – To my knowledge, it has not been shown that fixing the CO bias improves model OH biases. It is true that the low CO bias in the NH is well known, and the model OH bias is well known, but not that the OH bias is due to low CO. Please clarify your point here.

We did not intend to imply that the OH bias is exclusively due to the model underestimate in CO, merely that it could be a contributing factor, since it is the dominant sink on a global scale. Indeed, our own results suggest there must be additional issues as the OH bias exists in both hemispheres, despite the hemispheric asymmetry of the CO bias. We have revised the sentence to read (Page 9, Line 391-393):

> "This NH low bias in CO is a well-known problem in global chemistry models (Naik et al., 2013; Stein et al., 2014; Travis et al., 2020) and could be a contributing factor in the overestimate in OH, as CO is the dominant global OH sink."

Fig. 2 caption, please explain what you mean by "satellite lifetime?"

We have edited the text to clarify that we use data averaged from the beginning of available data for a given satellite (e.g. 2003 for AIRS) through 2018, when the simulation ended. The updated text reads:

> Figure 1: Tropospheric column CO (left), $H_2O_{(v)}$ (middle), and $NO_2$ (right) from MOPITT, AIRS, and OMI, respectively (top row), and MERRA2 GMI (middle row) for DJF. For the satellite retrievals and model, data are averaged over the time range described in the text for each instrument. The fractional difference between MERRA2 GMI and the satellite is shown in the bottom row.

Page 7, line 324 – Again it would be easier to read this if Fig. S3 was part of Fig. 2. It seems important enough to the discussion to warrant being in the main text.

As with the ATom 1 comparison, we originally chose to include the JJA satellite comparisons in the supplement to limit paper length and because the majority of the paper analysis focuses on DJF. But we understand your point so are now including the JJA satellite comparison in the main text as Figure 3.

Page 7, line 324 – You say that "overestimates of column CO, averaging 45%, in the SH corresponding with areas of biomass burning." Is that really true in SH winter (Fig. S3)?

You are correct. A more likely source of the CO overestimate in this region is a known model bias in isoprene, which would lead to an overestimate in *in situ* CO production from biogenic species. The text now reads (Page 10, 447-449):

> "These areas of high bias over South America likely result from the high bias in isoprene emissions, as discussed in Section 2.2, that would lead to unrealistically high *in situ* production of CO."

Page 8, line 341 – Are you re-calculating the OMI $NO_2$ columns using the model a priori to give you an apples to apples comparison, per Lamsal et al., 2014 (doi:10.5194/acp-14-11587-2014)?

We have not. We incorrectly indicated in the paper that we used version 3 of the OMI product, when we actually used version 4 (Lamsal, 2021). The *a priori* used for the OMI retrievals is from a GMI replay run using MERRA2 meteorology and the same emissions used for the simulation described here, and as such, according to Lok Lamsal, it is not necessary to re-calculate the OMI columns. We have updated the text and relevant references to reflect that we were using version 4 data. And, in response to comments from the other reviewer we have added the following paragraph discussing the satellite/model comparison (Page 9, Line 421-431):

> "For comparison of the satellite retrievals to MERRA2 GMI, we use monthly fields of the model variables output at the satellite overpass time. For CO, where averaging kernel and *a priori* information are available for the Level 3 MOPITT data, we convolve the model output with these variables so that direct comparison between satellite and model are possible. While shape factors and scattering weights for the OMI $NO_2$ retrieval are unavailable for the Level 3 data, shape factors for the OMI $NO_2$ retrieval are determined from a similar setup of the GEOSCCM model, also employing the GMI chemical mechanism and MERRA2 meteorology. Applying the satellite shape factors to the simulation discussed here would therefore not result in significant changes in the modeled $NO_2$. Finally, for AIRS $H_2O$, averaging kernel information was unavailable for the Level 3 data, so numerical comparisons between satellite and model should be regarded as more qualitative than quantitative."

Page 10, line 408 – You say, "The relative importance of the individual reactions is similar during neutral and La Niña years (not shown) and is in agreement with previous model studies (e.g. Spivakovsky et al., 2000)." I am confused since you show in Fig. 4 that ENSO is correlated with TCOH, mainly in the tropics. How can this be if the relative importance of R1-4 is the same during neutral and La Nina years?

While the relative importance of the individual reactions is the same during El Niño, La Niña, and neutral events (e.g., the $NO + HO_2$ reaction is always the dominant reaction for OH production in the tropical UFT, regardless of ENSO phase), the production rate from that reaction can still vary with ENSO phase. So, while the OH production rate from this reaction is lower over much of the tropics during an El Niño event than during other phases, OH production from this particular reaction is still larger than for the other reactions. Similar reasoning applies to the other atmospheric levels.

We have added the following text to help clarify this point (Page 13, Line 556-559):

> While the production rates along these pathways vary with the ENSO phase, as discussed in Sections 5.2 and 5.3, the relative importance of the individual reactions is similar during neutral and La Niña years (not shown) and is in agreement with previous model studies (e.g. Spivakovsky et al., 2000).

Page 10, line 413 – You do not use the term 'UFT' in the supplement (Fig. S4). Please be consistent in your terminology.

Thanks for pointing this out. This was left over from a previous set of terminology, and we missed updating this figure. We have updated the labels in the figure (now Fig. S2).

Page 12, line 456 – Does this mean that the variability shown in Fig 6a is driven mainly by changes in DJF as shown in Fig. 7a? If so, should Fig. 6 be shown seasonally similar to Fig. 7?

Figure 6a shows the anomalies in TCOH in DJF, while Figure 7a shows the spatial pattern of the first EOF of TCOH for DJF. Because the spatial pattern of the two figures is nearly identical (along with the high correlation between the MEI and temporal component of the first EOF), this indicates that the first EOF is likely ENSO-related for DJF. Similarly, agreement between the spatial pattern of the 1st EOF and El Niño anomaly plots of TCOH for the other

seasons would suggest that they are ENSO-related as well. We have added a supplementary figure (Fig. S5) showing the anomalies for the other seasons. As is expected from the $r^2$ values of the correlation of the MEI with temporal component of the EOFs, the spatial pattern of the first EOF for SON agrees more closely with the El Niño anomaly plot than do JJA and MAM which show lower correlation.

[Figure]

**Figure S2:** *Same as panel a of Figure 7 but for MAM (a), JJA (b), and SON (c).*

In addition, we have added the following clarification to the text (Page 15, Line 641-642): "Likewise, the spatial patterns of the first EOF of TCOH for these seasons are similar to the composite figures showing OH anomalies during El Niño (Fig. S5)."

Page 14, line 514 – I don't understand the plot of satellite $NO_2$ against the MEI for Fig. 8. You say that "OMI data are insufficient," but show a plot anyway. If you aren't going to talk about it, maybe don't bother showing it. If you are filtering out noise from the $NO_2$ retrieval, and this leaves you only with $NO_2$ over land and outflow regions, why do you show ocean values for $NO_2$ in Fig. 2?

We originally omitted this data because the correlations over the ocean demonstrate significantly more noise for OMI $NO_2$ than for the other species and are much more frequently below our significance level ($r > 0.5$). We have decided to take your suggestion and include the OMI $NO_2$ regression, however, because we do find that, despite these limitations, there is still information to be found in the regressions. We have updated the figure, shown below, and include the following text (Page 17, Line 713-720):

> Correlations between OMI $NO_2$ and the MEI suggest similar relationships as found in the MERRA2 GMI simulation, although the correlations are not as robust as for the other satellite variables examined here. This is likely because tropospheric $NO_2$ column over the ocean are frequently at or below the instrumental average noise ($5 \times 10^{14}$ molecules/cm$^2$). As with the simulation, OMI suggests broad regions of anti-correlation between ENSO and $NO_2$ in the equatorial Pacific and Gulf of Alaska as well as a region of positive correlation in the extra-tropical NH Pacific. These results demonstrate that, with enough temporal and spatial averaging, OMI is capable of capturing the variability of tropospheric $NO_2$ even in remote regions with low concentrations.

[Figure]

Figure S6 – Should the caption say, "model" $NO_2$ to differentiate from satellite?

We have removed this figure based on the suggestion of the other reviewer.

Page 14, line 535 – Is there a plot somewhere for the sentence "Changes in the PBL during La Nina…?"
We have added a reference to Fig. 10b to the paper. We have also updated the text to emphasize that these are localized changes. (Page 18, Line 760).

Pate 14, line 547 – You might want to consider the mass-weighted concentration of OH more consistently throughout the paper, and possibly the methane lifetime as well. This might provide an additional perspective on how different climate indices change the atmospheric oxidation capacity and address the variability in methane lifetime across models that you describe in the introduction.

We have changed most of the references to percent changes in OH concentration to changes in mass weighted OH concentration when discussing global means. Although we do make occasional references to global or regional changes in OH, our aim is to focus more on the spatial relationship between OH and climate modes. We have included a brief discussion of methane lifetime, which we explain more fully in response to your comment regarding page 24, Line 832.

Page 15, line 545 – According to Table 1, the $r^2$ with ENSO in JJA is less than 0.25, this should be pointed out.

The 0.25 correlation is actually for the tropospheric column, while the line you reference is discussing the PBL, where correlation is much lower ($r^2 = 0.07$). We have updated the text in the EOF discussion in both the column (Page 15, Line 535-537):

> In JJA, ENSO influence on OH is much weaker, with a correlation between the 1st EOF and TCOH of $r^2 = 0.25$, consistent with the seasonal cycle of ENSO.

And the PBL discussion (Page 18, Line 771-773):

> In general, the $r^2$ with ENSO is 0.5 or higher and the mode contributes approximately 10% of the total variance, although correlation in JJA ($r^2 = .07$) is negligible.

Page 15, line 551 – Europe looks like it has decreases not increases, but it is hard to tell with the way the plot is centered. Please double check.

We've updated the text to reflect that the sign of the EOF is not constant over Europe, as it is positively signed over portions over the UK, Scandinavia, and other parts of Northern Europe, and negatively signed over other portions of the continent. We now say (Page 18, Line 780) "…net emissions reductions (the United States, portions of Europe, …)".

Page 17, line 588 – Do you mean "These increases in jO1D", not "These increases in O1D?" Also, does the model show that decreased stratospheric ozone is the driver behind the increase in photolysis?

Yes, we meant JO1D. The model does show decreases in total stratospheric column $O_3$ during El Niño events, which we now note in the paper. Showing an explicit linkage between the decrease in column $O_3$ and JO1D is beyond the scope of the work. We have refined our language to indicate more uncertainty around the linkage (Page 20, Line 841-844):

> The MERRA2 GMI simulation shows reduction in total stratospheric column $O_3$ of 2-5% in the tropics during El Niño, consistent with previous work (e.g., Randel et al., 2009), which could contribute to the increase in $JO^1D$, although more work is needed to establish this link.

Page 17, line 621 – It is surprising to me that if stratospheric ozone impacted jO1D in the PBL, it wouldn't also be important in the UFT. Please clarify.

There is positive correlation between $JO^1D$ and ENSO in the UFT, as demonstrated by the figure below, which would suggest increased OH production from this reaction in the upper troposphere. The overall importance of the $O^1D + H_2O$ reaction, however, is much lower in the tropical UFT, where it contributes only about 10% of total OH production, than it is in the PBL, where it contributes 70% or greater of total OH production. Because of this minor importance in the UFT, any changes in production from the $O^1D + H_2O$ reaction are dwarfed by changes from secondary production by the $NO + HO_2$ reaction, which comprises ~80% of total OH production. We have added the following text to the paper to clarify this point (Page 20, Line 873-889):

> While $JO^1D$ does increase in the UFT during El Niño events, as does production from the $O^1D + H_2O$ reaction in some regions, the relatively small contribution of this reaction to the total OH production in the UFT (Fig. **Error! Reference source not found.**a) does not significantly perturb OH in this layer.

[Figure]

Correlation between $JO^1D$ in the UFT with the MEI.

Page 18, line 638 – I don't think the reference to Fig. S7 is correct, did you mean to refer to a different figure?

Based on the suggestion of the other reviewer, he have removed this figure.

Page 18, line 645 – H2O(v) is Fig. 8a, not Fig. 8c.

We have made this change.

Page 18, line 646 – It seems that there is a figure, Fig. 16, that you should refer to here in your discussion of lightning.

We have added a reference to what is now Figure 17 on Page 22, Line 919.

Page 19, line 652 – I don't understand your discussion of Turner et al. (2018). If I understand Turner et al. Fig. 4, correctly, there is more lightning during La Nina, but less during El Nino and thus La Nina is positively correlated

with increases in OH. If I understand your Fig. 15 correctly, if NO goes down as MEI goes up, then NO goes up as the MEI goes down, meaning more lightning during a La Nina. This actually agrees with the findings of Turner et al. Please clarify if I am missing something here. If I am wrong, you will definitely need to better clarify your statement about biomass burning and be more specific about how interannual variability in emissions could have changed your results vs. Turner et al., 2018. For example, maybe you need a correlation plot of biomass burning emissions with MEI.

We have overhauled the discussion of Turner *et al* as we agree it was not clearly written. We had intended to convey the idea that additional mechanisms, besides lightning NO production, likely lead to the modeled relationship between $NO_y$ species and ENSO in the equatorial Pacific, where the model shows a narrow equatorial band where NO positively correlates with the MEI (Fig. 17) that overlaps spatially with the region werhe overall NO (as well as other $NO_y$ species) is anti-correlated (Fig. 16). Outside of the equatorial region, both lightning flash rates and the overall NO concentrations are anti-correlated with the MEI, suggesting decreased NO during El Niño, which is consistent with the decreased OH concentrations, indicating increased NO during La Niña, as shown to be the case when on integrates the LIS/OTD flash count product over the entire tropical region available (Turner et al., 2018). We have updated this discussion as follows (Page 23, Lines 947-965):

> This tri-pole correlation pattern between MEI and lightning, evident in both the satellite and model (Fig. 17) is in contrast to the relationship with NO (Fig. 16a) and other reactive nitrogen ($NO_y$) species in the UFT. While the anti-correlation in NO is consistent with the changes in lightning NO emissions in some regions, in the equatorial Pacific band, NO decreases during El Niño events despite an increase in lightning NO emissions. This apparent discrepancy occurs because even though lightning NO increases by 100% or more over the equatorial Pacific during El Niño events in the model, the absolute difference is orders of magnitude lower than the accompanying changes over land. We conclude that the resulting NO perturbations over the equatorial Pacific latitudes are dominated by mechanism other than the local lightning response, such as changes in the Walker Circulation and the associated transport of air originating over the continents. This mechanism is supported by the similar regression pattern of longer-lived species, such as $HNO_3$ (Fig. **Error! Reference source not found.**c) and PAN (not shown), to NO in the UFT supports this idea, showing that transport of reactive nitrogen from other source regions, particularly lightning over South America, is likely reduced during El Niño events.
>
> Our findings are broadly consistent with Turner et al. (2018), who found that increases in lightning NO emissions drive increases in OH during La Niña and, conversely, decreases in lightning NO emissions lead to OH decreases during El Niño. The results presented here suggest that in addition to this influence of lightning locally, other mechanisms, such as atmospheric transport of $NO_y$ species, also likely contribute to the relationship between ENSO and OH in the equatorial Pacific.

Page 19, line 667 – It seems that part of the argument is given here. Again, I would suggest that a biomass burning correlation plot or something similar would strengthen your argument. It does seem unlikely though that biomass burning would have such a large impact on the UFT, this definitely needs more discussion.

We agree that further analysis of the role of biomass burning would be helpful. Unfortunately, no biomass burning tracers, such as HCN or tagged CO, were output by the model, so the only biomass burning specific data available are the emissions. So, while we do think this is an interesting question, determining the relative impact of biomass burning on UFT OH during El Niño would require substantial new modeling work that is beyond the scope of this paper.

Page 21, line 740 – What is going on such that EMAC has no correlation in the column, but a very strong correlation in the UFT?

We have avoided addressing the causes of inter-model differences as well as the drivers in the individual CCMI models because a model intercomparison is beyond the scope of this paper. Other work has already made strides in exploring inter-model differences in OH in the CCMI models (e.g. Nicely, 2019 and Zhao, 2019) and we refer the reader to those (Page 6, Line 287). That being said, it's not inconsistent that the first EOF of OH in the UFT is correlated with the MEI and the first EOF of TCOH is not. Because different reactions control OH production at different levels in the atmosphere, it is possible that, in EMAC, ENSO affects OH production from the dominant NO + $HO_2$ reaction in the UFT while ENSO does not affect OH production reactions in the other atmospheric levels. In

addition, the UFT is only a small portion of the tropospheric column, spanning from 300 hPa to the tropopause, so changes in this level might not significantly perturb column amounts. We have added the following sentence to the paper (Page 25, Line 1108-1111):

> While further work is needed to understand the cause of the relationship between OH and ENSO in the UFT in EMAC, results from MERRA2 GMI suggest a role for changes in production via the NO + HO$_2$ reaction.

Figure 18 – While you focus on the agreement in your discussion of this figure, the discrepancies are actually quite large considering that all 5 models do not agree on the relationship of ENSO to TCOH in most cases. While you say it is beyond the scope of the paper on Page 22, line 748, given your analysis on the drivers of the ENSO to TCOH relationship from MERRA2 GMI, can you suggest areas of focus that might help us understand the huge model to model variability on OH and methane lifetime that you discuss in the introduction (e.g., Voulgarakis et al., 2013;Nicely et al.,;Zhao et al., 2019)?

As discussed above, we do not want to delve too deeply into the world of inter-model comparisons, but your suggestion here does make sense.  We have added the following text to the end of Section 5.5, discussing possible sources of disagreement among the models (Page 26, Line 1128-1132):

> Given the results from the MERRA2 GMI analysis, investigating ENSO-related changes in UFT NO, both from lightning and transport, could provide insight into these inter-model differences. Further, Nicely et al. (2020) showed that JO[1]D was the largest driver in differences in the methane lifetime in the CCMI models, suggesting the potential importance of this variable in inter-model differences in the OH-ENSO relationship in the PBL and lower troposphere.

Page 22, line 767 – It seems like Figure S13 is more appropriate for the main text.

We have moved this figure to the main text as Figure 20.

Page 22, line 768 – Here you do talk about mass-weighted OH. I will just reiterate that it could be extremely useful to calculate changes to mass-weighted OH, or even better, changes to the methane lifetime in El Nino vs. La Nina.

As suggested above, we have increased the inclusion of the mass-weighted OH metric.  We discuss the relationship between ENSO and methane lifetime in a response below.

Page 22, line 789 – Can you check the model output during JJA to see whether there is really no interannual variability? It does seem more likely that the model is not accurately capturing the chemical variability within the monsoon anticyclone.

We do not find significant interannual variability in OH above India during the monsoon season, but a full evaluation of this would require defining the location and bounds of the monsoon anticyclone for each year, which we feel is beyond the scope of this work.  We have omitted the first part of the sentence, and now simply say (Page 27, Line 1185-1186): "The lack of correlation demonstrated here suggests that the model is not accurately capturing the chemical variability within the monsoon anticyclone".

Page 23, line 819 – Could this not also be that most models fail to capture the mechanism behind the monsoon impact on OH? If it is due to convectively lofting pollution above the monsoon clouds, then it seems very likely that all models have difficulty with this. Could you look to see whether your model has expected elevated levels of pollution during the monsoon in the UFT that might drive increased OH as described by Lelieveld et al., 2018?

Yes, the models could be missing important transport mechanisms and have included this in the text, which now reads (Page 28, Line 1234-1236) "…the lack of correlation among the models suggests either that those changes are not highly variable from year to year or that not all models capture the mechanisms behind monsoon influence on OH, such as convective lofting of OH precursors."  To examine this more deeply would require a more in-depth analysis (e.g., defining monsoonal circulation) that will further lengthen the paper.

Page 24, line 832 – Could you say something as well about the average decrease in the methane lifetime (using the methane concentrations from the model)?

We have now included a discussion of methane lifetime, but in section 5.4, where we discuss ENSO-related changes in OH in the LFT and MFT.  As mentioned earlier, globally averaged changes in mass-weighted tropospheric OH are only ~2% during El Niño events, when compared to neutral events, which corresponds to an approximately 1%

increase in methane lifetime during El Niño events. While we demonstrate throughout the paper that ENSO drives OH variability over much of the globe, when viewed with global annual mean metrics, the variability effectively averages out. So, we do not expand on the $CH_4$ lifetime beyond the following text (Page 24, Lines 1057-1063):

> "The comparatively smaller changes in LFT OH during El Niño events limit the effect of ENSO on the interannual variability of the $CH_4$ lifetime. Global mean, mass-weighted tropospheric OH decreases by 2.2% during El Niño events, corresponding to only a 1% decrease in the $CH_4$ lifetime. While changes in OH concentration are most pronounced in the UFT and PBL, $CH_4$ lifetime is mostly dictated by OH in the LFT due to the temperature dependence of the OH + $CH_4$ reaction rate. This limited effect on $CH_4$ lifetime highlights the importance of investigating the spatial OH variability as global mean metrics can obscure important year-to-year changes."

Page 24, line 844 – This statement about lightning emissions is very important, and definitely needs more support in the prior text. I don't think that "increased convective lofting of low NO air from near the surface and advection of air with lower reactive nitrogen than during neutral years" has been well supported.

We have updated the text to reflect the clarifications we discussed earlier in reference to our characterization of Turner et al and the relationship between lightning NO and transport of NO in the tropical Pacific. The text now reads (Page 29, Line 1276-1281):

> In much of the region, decreases in lightning NO production correspond to decreases in total NO, and thus OH. In the equatorial region, however, increases in lightning NO production are offset by other processes, potentially including transport due to changes in the Walker Circulation. Further work is needed to determine the relative importance of these two factors in controlling OH in the region during El Niño and La Niña events.

Technical Corrections
Page 7, line 288 – "a NMB", not "an NMB"
Page 15, line 546 – I think it should be "ENSO-related".

We have made both of these corrections.